# Residual2Vec: Debiasing graph embedding with random graphs

**Sadamori Kojaku**[1], **Jisung Yoon**[2], **Isabel Constantino**[3], **Yong-Yeol Ahn**[1,3,4]

[1]Center for Complex Networks and
Systems Research, Luddy School of
Informatics, Computing and
Engineering, Indiana University, USA

[2]Department of Industrial
Management and Engineering
Pohang University of Science and
Technology, South Korea

[3]Indiana University
Network Science Institute
Indiana University, USA

[4]Connection Science
Massachusetts Institute of
Technology, USA

{skojaku, imconsta, yyahn}@iu.edu, jisung.yoon92@gmail.com

## Abstract

Graph embedding maps a graph into a convenient vector-space representation for graph analysis and machine learning applications. Many graph embedding methods hinge on a sampling of context nodes based on random walks. However, random walks can be a biased sampler due to the structural properties of graphs. Most notably, random walks are biased by the degree of each node, where a node is sampled proportionally to its degree. The implication of such biases has not been clear, particularly in the context of graph representation learning. Here, we investigate the impact of the random walks' bias on graph embedding and propose *residual2vec*, a general graph embedding method that can debias various structural biases in graphs by using random graphs. We demonstrate that this debiasing not only improves link prediction and clustering performance but also allows us to explicitly model salient structural properties in graph embedding.

## 1 Introduction

On average, your friends tend to be more popular than you. This is a mathematical necessity known as the *friendship paradox*, which arises due to a sampling bias, i.e., popular people have many friends and thus are likely to be on your friend list [1]. Beyond being a fun trivia, the friendship paradox is a fundamental property of graphs: following an edge is a biased sampling that preferentially samples nodes based on nodes' degree (i.e., the number of neighbors). The fact that random walk is used as the default sampling paradigm across many graph embedding methods raises important questions: what are the implications of this sampling bias in graph embedding? If it is undesirable, how can we debias it?

Graph embedding maps a graph into a dense vector representation, enabling a direct application of many machine learning algorithms to graph analysis [2]. A widely used framework is to turn a graph into a "sentence of nodes" and then feed the sentence to `word2vec` [3–6]. A crucial difference from word embedding is that, rather than using given

35th Conference on Neural Information Processing Systems (NeurIPS 2021).

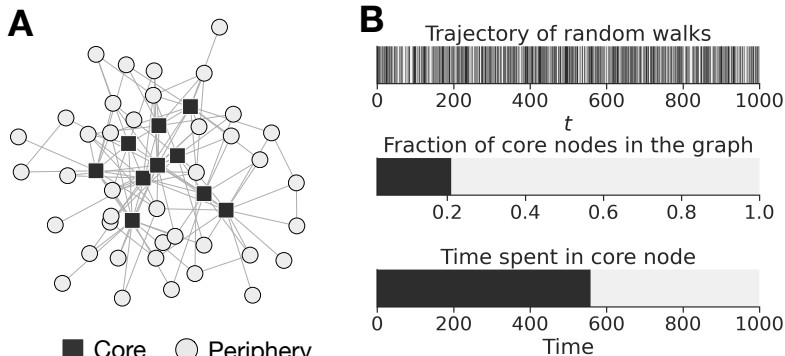

Figure 1: Random walks have a strong preference towards hubs. (**A**) A toy graph generated by a stochastic block model with a core-periphery structure, where core nodes have more neighbors than peripheral nodes [18]. (**B**) Random walkers preferentially visit nodes with many neighbors, generating a trajectory that overrepresents the core nodes.

sentences, graph embedding methods generate synthetic "sentences" from a given graph. In other words, the generation of synthetic "sentences" in graph is an implicit modeling decision [7], which most graph embedding methods take for granted. A common approach for generating sentences from a graph is based on random walks, which randomly traverse nodes by following edges. The friendship paradox comes into play when a walker follows an edge (e.g., friendship tie): it is more likely to visit a node with many neighbors (e.g., popular individual). As an example, consider a graph with a core-periphery structure, where core nodes have more neighbors than periphery (Fig. 1A). Although core nodes are the minority, they become the majority in the sentences generated by random walks (Fig. 1B). This is because core nodes have more neighbors than periphery and thus are likely to be a neighbor of other nodes, which is a manifestation of the friendship paradox. Then, how does the sampling bias affect the embedding?

Previous approaches to mitigate the degree bias in embedding are based on modifying random walks or the post-transformation of the embedding [8–16]. Here we show that word2vec by itself has an implicit bias arising from the optimization algorithm—skip-gram negative sampling (SGNS)—which happens to negate the bias due to the friendship paradox. To leverage this debiasing feature further, we propose a more general framework, `residual2vec`, that can also compensate for other systematic biases in random walks. We show that `residual2vec` performs better than conventional embedding methods in link prediction and community detection tasks. Using a citation graph of 260k journals, we demonstrate that the biases from random walks overshadow the salient features of graphs. By removing the bias, `residual2vec` better captures the characteristics of journals such as the impact factor and journal subject. The python code of residual2vec is available at GitHub [17].

## 2 Built-in debiasing feature of SGNS word2vec

### 2.1 Background: SGNS word2vec

Consider a sentence of words $(x_1, x_2, x_3, \dots)$ composed of $N$ unique words. `word2vec` associates the $t$th word $x_t$ with words in its surrounding $x_{t-T}, \dots, x_{t-1}, x_{t+1}, \dots, x_{t+T}$, which are referred to as context words, determined by a prescribed window size $T$. For a center-context word pair $(i, j)$, `word2vec` models conditional probability

$$P_{\text{w2v}}(j|i) = \frac{\exp(\boldsymbol{u}_i^\top \boldsymbol{v}_j)}{\sum_{j'=1}^{N} \exp(\boldsymbol{u}_i^\top \boldsymbol{v}_{j'})}, \tag{1}$$

where $\boldsymbol{u}_i, \boldsymbol{v}_i \in \mathbb{R}^{K \times 1}$ are *embedding vectors* representing word $i$ as center and context words, respectively, and $K$ is the embedding dimension. An approach to fit $P_{\text{w2v}}$ is the maximum likelihood estimation, which is computationally expensive because $P_{\text{w2v}}$ involves the sum

over all words. Alternatively, several heuristics have been proposed, among which *negative sampling* is the most widely used [3, 4, 6].

Negative sampling trains `word2vec` as follows. Given a sentence, a center-context word pair $(i, j)$ is sampled and labeled as $Y_j = 1$. Additionally, one samples $k$ random word $\ell$ as candidate context words from a noise distribution $p_0(\ell)$, and then labels $(i, \ell)$ as $Y_\ell = 0$. In general, a popular choice of the noise distribution $p_0(\ell)$ is based on word frequency, i.e., $p_0(\ell) \propto P_d(\ell)^\gamma$, where $P_d(\ell)$ is the fraction of word $\ell$ in the given sentence, and $\gamma$ is a hyper-parameter. Negative sampling trains $\boldsymbol{u}_i$ and $\boldsymbol{v}_j$ such that its label $Y_j$ is well predicted by a logistic regression model

$$P_{\mathrm{NS}}(Y_j = 1; \boldsymbol{u}_i, \boldsymbol{v}_j) = \frac{1}{1 + \exp(-\boldsymbol{u}_i^\top \boldsymbol{v}_j)}, \tag{2}$$

by maximizing its log-likelihood.

## 2.2 Implicit debiasing by negative sampling

Negative sampling efficiently produces a good representation [6]. An often overlooked fact is that negative sampling is a simplified version of Noise Contrastive Estimation (NCE) [19, 20], and this simplification biases the model estimation. In the following, we show that this estimation bias gives rise to a built-in debiasing feature of SGNS word2vec.

**Noise contrastive estimation** NCE is a generic estimator for probability model $P_m$ of the form [19]:

$$P_m(x) = \frac{f(x; \theta)}{\sum_{x' \in \mathcal{X}} f(x'; \theta)}, \tag{3}$$

where $f$ is a non-negative function of data $x$ in the set $\mathcal{X}$ of all possible values of $x$. `word2vec` (Eq. (1)) is a special case of $P_m$, where $f(x) = \exp(x)$ and $x = u_i^\top v_j$. NCE estimates $P_m$ by solving the same task as negative sampling—classifying a positive example and $k$ randomly sampled negative examples using logistic regression—but based on a Bayesian framework [19, 20]. Specifically, as prior knowledge, we know that 1 in $1 + k$ pairs are taken from the given data, which can be expressed as prior probabilities [19, 20]:

$$P(Y_j = 1) = \frac{1}{k + 1}, \quad P(Y_j = 0) = \frac{k}{k + 1}. \tag{4}$$

Assuming that the given data is generated from $P_m$, the positive example $(Y_j = 1)$ and the negative examples $(Y_j = 0)$ are sampled from $P_m$ and $p_0(j)$, respectively [19, 20], i.e.,

$$P(j|Y_j = 1) = P_m(\boldsymbol{u}_i^\top \boldsymbol{v}_j), \quad P(j|Y_j = 0) = p_0(j). \tag{5}$$

Substituting Eqs. (4) and (5) into the Bayes rule yields the posterior probability for $Y_j$ given an example $j$ [19, 20]:

$$P_{\mathrm{NCE}}(Y_j = 1|j) = \frac{P(j|Y_j = 1) P(Y_j = 1)}{\sum_{y \in \{0,1\}} P(j|Y_j = y) P(Y_j = y)} = \frac{P_m(\boldsymbol{u}_i^\top \boldsymbol{v}_j)}{P_m(\boldsymbol{u}_i^\top \boldsymbol{v}_j) + k p_0(j)}, \tag{6}$$

which can be rewritten with a sigmoid function as

$$P_{\mathrm{NCE}}(Y_j = 1|j) = \frac{1}{1 + k p_0(j) / P_m(\boldsymbol{u}_i^\top \boldsymbol{v}_j)} = \frac{1}{1 + \exp\left[-\ln f(\boldsymbol{u}_i^\top \boldsymbol{v}_j) + \ln p_0(j) + c\right]}, \tag{7}$$

where $c = \ln k + \ln \sum_{x' \in \mathcal{X}} f(x')$ is a constant. NCE learns $P_m$ by the logistic regression based on Eq. (7). The key feature of NCE is that it is an asymptomatically unbiased estimator of $P_m$ whose bias goes to zero as the number of training examples goes to infinity [19].

**Estimation bias of negative sampling** In the original paper of word2vec [6], the authors simplified NCE into negative sampling by dropping $\ln p_0(j) + c$ in Eq. (7) because it reduced the computation and yielded a good word embedding. In the following, we show the impact of this simplification on the final embedding.

We rewrite $P_{\mathrm{NS}}$ (i.e., Eq. (1)) in the form of $P_{\mathrm{NCE}}$ (i.e., Eq. (7)) as

$$P_{\mathrm{NS}}(Y_j = 1; \boldsymbol{u}_i, \boldsymbol{v}_j) = \frac{1}{1 + \exp(-\boldsymbol{u}_i^\top \boldsymbol{v}_j)} = \frac{1}{1 + \exp\left[-\left(\boldsymbol{u}_i^\top \boldsymbol{v}_j + \ln p_0(j) + c\right) + \ln p_0(j) + c\right]}$$

$$= \frac{1}{1 + \exp\left[-\ln f(\boldsymbol{u}_i^\top \boldsymbol{v}_j) + \ln p_0(j) + c\right]}, \tag{8}$$

Equation (8) makes clear the relationship between negative sampling and NCE: negative sampling is the NCE with $f(\boldsymbol{u}_i^\top \boldsymbol{v}_j) = \exp\left(\boldsymbol{u}_i^\top \boldsymbol{v}_j + \ln p_0(j) + c\right)$ and noise distribution $p_0$ [21]. Bearing in mind that NCE is the asymptomatically unbiased estimator of Eq. (3) and substituting $f(\boldsymbol{u}_i^\top \boldsymbol{v}_j)$ into Eq. (3), we show that SGNS word2vec is an asymptomatically unbiased estimator for probability model:

$$P_{\mathrm{w2v}}^{\mathrm{SGNS}}(j \mid i) = \frac{p_0(j) \exp(\boldsymbol{u}_i^\top \boldsymbol{v}_j)}{Z_i'}, \quad \text{where } Z_i' := \sum_{j'=1}^{N} p_0(j') \exp(\boldsymbol{u}_i^\top \boldsymbol{v}_{j'}). \tag{9}$$

Equation (9) clarifies the role of noise distribution $p_0$. Noise probability $p_0$ serves as a baseline for $P_{\mathrm{w2v}}^{\mathrm{SGNS}}$, and word similarity $\boldsymbol{u}_i^\top \boldsymbol{v}_j$ represents the *deviation* from $p_0(j)$, or equivalently the characteristics of words *not* captured in $p_0(j)$. Notably, baseline $p_0(j) \propto P_{\mathrm{d}}(j)^\gamma$ is determined by word frequency $X_{\mathrm{d}}(j)$ and thus negates the word frequency bias. This realization—that we can explicitly use a noise distribution to obtain "residual" information—is the motivation for our method, *residual2vec*.

## 3    Residual2vec graph embedding

We assume that the given graph is undirected and weighted, although our results can be generalized to directed graphs (see Supplementary Information). We allow multi-edges (i.e., multiple edges between the same node pair) and self-loops, and consider unweighted graphs as weighted graphs with all edge weight set to one [22, 23].

### 3.1    Model

The presence of $p_0(j)$ effectively negates the bias in random walks due to degree. This bias dictates that, for a sufficiently long trajectory of random walks in undirected graphs, the frequency $P_{\mathrm{d}}(j)$ of node $j$ is proportional to degree $d_j$ (i.e., the number of neighbors) irrespective of the graph structure [24]. Now, if we set $\gamma = 1$, baseline $p_0(j) = P_{\mathrm{d}}(j)$ matches exactly with the node frequency in the trajectory, negating the bias due to degree. But, we are free to choose any $p_0$. This consideration leads us to *residual2vec* model:

$$P_{\mathrm{r2v}}(j \mid i) = \frac{P_0(j \mid i) \exp(\boldsymbol{u}_i^\top \boldsymbol{v}_j)}{Z_i'}, \tag{10}$$

where we explicitly model baseline transition probability denoted by $P_0(j \mid i)$. In doing so, we can obtain the *residual* information that is not captured in $P_0$. Figure 2 shows the framework of `residual2vec`. To negate a bias, we consider "null" graphs, where edges are randomized while keeping the property inducing the bias intact [22, 23]. Then, we compute $P_0$ either analytically or by running random walks in the null graphs. The $P_0$ is then used as the noise distribution to train SGNS `word2vec`.

**Random graph models**  Among many models for random graph [25–31], here we focus on the degree-corrected stochastic block model (dcSBM), which can be reduced to many fundamental random graph models with certain parameter choices [28]. With the dcSBM, one partitions nodes into $B$ groups and randomizes edges while preserving (i) the degree of each node, and (ii) the number of inter-/intra-group edges. Preserving such group connectivity is useful to negate biases arising from less relevant group structure such as bipartite and multilayer structures. The dcSBM can be mapped to many canonical ensembles that preserve the expectation of structural properties. In fact, when $B = 1$, the dcSBM is reduced to the soft configuration model that preserves the degree of each node on average, with self-loops

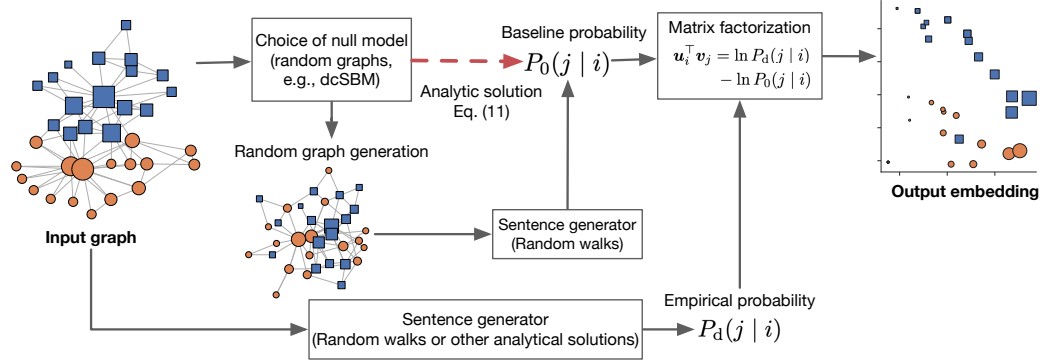

Figure 2: `residual2vec` framework. We first choose an explicit "null" model (e.g., dcSBM random graph). By running random walks in the input and random graphs, "sentences of nodes" are generated and then compared to extract the residual information not in the random graphs. The final embedding encodes the residual information. Probabilities $P_d(j \mid i)$ and $P_0(j \mid i)$ can be computed analytically in many cases. For instance, we use the analytical solution for dcSBM (i.e., $P_d(j \mid i)$ and $P_0(j \mid i)$) instead of simulating actual random walks.

Table 1: Baseline probability $P_0(j \mid i)$ for the dcSBM along with its special cases.

| | Canonical random graph models | | |
|---|---|---|---|
| | Erdős-Rényi model for multigraphs [25] | The soft configuration model [31] | dcSBM [28] |
| $P_0(j \mid i)$ | $1/N$ | $d_j / \sum_{\ell=1}^N d_\ell$ | $d_j D_{g_j}^{-1} \left( T^{-1} \sum_{t=1}^T \mathbf{P}_{\text{SBM}}^t \right)_{g_i,g_j}$ |
| Algorithm | `DeepWalk` [3] | `node2vec` ($\gamma = 0.75$) [4] 
 `NetMF` ($\gamma = 1$) [32] | |

and multi-edges allowed [31]. Furthermore, by setting $B = 1$ and $d_i = $ constant, the dcSBM is reduced to the Erdős-Rényi model for multigraphs that preserves the number of edges on average, with self-loops and multi-edges allowed. In the dcSBM, the edge weights follow a Poisson distribution and thus take integer values.

Suppose that the nodes in the given graph have $B$ discrete labels (e.g., gender), and we want to remove the structural bias associated with the labels. If no such label is available, all nodes are considered to have the same label (i.e., $B = 1$). We fit the dcSBM with $B$ groups, where each group consists of the nodes with the same label. The dcSBM generates random graphs that preserve the number of edges within and between the groups (e.g., assortativity by gender types). We can calculate $P_0(j \mid i)$ without explicitly generating random graphs (Supplementary Information)

$$P_0(j \mid i) = \frac{d_j}{D_{g_j}} \left( \frac{1}{T} \sum_{t=1}^T \mathbf{P}_{\text{SBM}}^t \right)_{g_i,g_j}, \tag{11}$$

where node $j$ has degree $d_j$ and belongs to group $g_j$, $D_g = \sum_{\ell=1}^N d_\ell \delta(g_\ell, g)$, and $\delta$ is Kronecker delta. The entry $P_{g,g'}^{\text{SBM}}$ of matrix $\mathbf{P}_{\text{SBM}} = (P_{g,g'}^{\text{SBM}}) \in \mathbb{R}^{B \times B}$ is the fraction of edges to group $g'$ in $D_g$. Table 1 lists $P_0(j \mid i)$ for the special classes of the dcSBM. See Supplementary Information for the step-by-step derivation.

**Other graph embedding as special cases of residual2vec** `residual2vec` can be considered as a general framework to understand structural graph embedding methods because many existing graph embedding methods are special cases of `residual2vec`. `node2vec` and `NetMF` use SGNS `word2vec` with $p_0(j) \propto P_d(j)^\gamma$. This $p_0(j)$ is equivalent to the baseline for the soft configuration model [31], where each node $i$ has degree $d_i^\gamma$. `DeepWalk` is also based on `word2vec` but trained with an unbiased estimator (i.e., the hierarchical softmax).

Because negative sampling with $p_0(j) = 1/N$ is unbiased [19, 20], `DeepWalk` is equivalent to `residual2vec` with the Erdős-Rényi random graphs for multigraphs.

## 3.2 Residual2vec as matrix factorization

Many structural graph embedding methods implicitly factorize a matrix to find embeddings [32]. `residual2vec` can also be described as factorizing a matrix $\mathbf{R}$ which captures residual pointwise mutual information. Just like a $K$th order polynomial function can be fit to $K$ points without errors, we can fit `word2vec` to the given data without errors when the embedding dimension $K$ is equal to the number of unique words $N$ [30]. In other words, $P_{\mathrm{d}}(j \mid i) = P_{\mathrm{r2v}}(j \mid i)$, $\forall i, j$ if $K = N$. By substituting $P_{\mathrm{r2v}}(j \mid i) = P_0(j \mid i) \exp(\boldsymbol{u}_i^\top \boldsymbol{v}_j)/Z_i'$, we obtain

$$P_{\mathrm{d}}(j \mid i) = \frac{P_{\mathrm{d}}(j \mid i) \exp\left(\boldsymbol{u}_i^\top \boldsymbol{v}_j + \ln P_0(j \mid i) - \ln P_{\mathrm{d}}(j \mid i)\right)}{\sum_{j'=1}^{N} P_{\mathrm{d}}(j' \mid i) \exp\left(\boldsymbol{u}_i^\top \boldsymbol{v}_{j'} + \ln P_0(j' \mid i) - \ln P_{\mathrm{d}}(j' \mid i)\right)}. \tag{12}$$

The equality holds if $\exp(\boldsymbol{u}_i^\top \boldsymbol{v}_j + \ln P_0(j \mid i) - \ln P_{\mathrm{d}}(j \mid i)) = c_i$ for all $i$ and $j$, where $c_i$ is a constant. The solution is not unique because $c_i$ can be any real value. We choose $c_i = 1$ to obtain a solution in the simplest form, yielding matrix $\mathbf{R}$ that `residual2vec` factorizes:

$$R_{ij} := \boldsymbol{u}_i^\top \boldsymbol{v}_j = \ln P_{\mathrm{d}}(j \mid i) - \ln P_0(j \mid i). \tag{13}$$

Matrix $\mathbf{R}$ has an information-theoretic interpretation. We rewrite

$$R_{ij} = \ln \frac{P_{\mathrm{d}}(i,j)}{P_{\mathrm{d}}(i)} - \ln \frac{P_0(i,j)}{P_0(i)} = \ln \frac{P_{\mathrm{d}}(i,j)}{P_{\mathrm{d}}(i)P_{\mathrm{d}}(j)} - \ln \frac{P_0(i,j)}{P_0(i)P_0(j)} + \ln P_{\mathrm{d}}(j) - \ln P_0(j). \tag{14}$$

The dcSBM preserves the degree of each node and thus has the same the degree bias with the given graph, i.e., $P_{\mathrm{d}}(i) = P_0(i)$ (Supplementary Information), which leads

$$R_{ij} = \mathrm{PMI}_{\sim P_{\mathrm{d}}}(i,j) - \mathrm{PMI}_{\sim P_0}(i,j), \text{ where } \mathrm{PMI}_{\sim P}(i,j) = \ln \frac{P(i,j)}{P(i)P(j)}. \tag{15}$$

$\mathrm{PMI}_{\sim P}(i,j)$ is the pointwise mutual information that measures the correlation between center $i$ and context $j$ under joint distribution $P(i,j)$, i.e., $\mathrm{PMI}_{\sim P}(i,j) = 0$ if $i$ and $j$ appear independently, and $\mathrm{PMI}_{\sim P}(i,j) > 0$ otherwise. In sum, $R_{ij}$ reflects residual pointwise mutual information of $i$ and $j$ from the null model.

## 3.3 Efficient matrix factorization

Although we assume that $N = K$ above, in practice, we want to find a compact vector representation (i.e., $K \ll N$) that still yields a good approximation [30, 32]. There are several computational challenges in factorizing $\mathbf{R}$. First, $\mathbf{R}$ is ill-defined for any node pair $(i, j)$ that never appears because $R_{ij} = \ln 0 = -\infty$. Second, $\mathbf{R}$ is often a dense matrix with $\mathcal{O}(N^2)$ space complexity. For these issues, a common remedy is a truncation [30, 32]:

$$\tilde{R}_{ij} := \max(R_{ij}, 0). \tag{16}$$

This truncation discards negative node associations ($R_{ij} < 0$) while keeping the positive associations ($R_{ij} > 0$) based on the idea that negative associations are common, and thus are less informative [32]. In both word and graph embeddings, the truncation substantially reduces the computation cost of the matrix factorization [30, 32].

We factorize $\tilde{\mathbf{R}} = (\tilde{R}_{ij})$ into embedding vectors $\boldsymbol{u}_i$ and $\boldsymbol{v}_j$ such that $\boldsymbol{u}_i^\top \boldsymbol{v}_j \simeq \tilde{R}_{ij}$ by using the truncated singular value decomposition (SVD). Specifically, we factorize $\tilde{\mathbf{R}}$ by $\tilde{\mathbf{R}} = \boldsymbol{\Phi}_{\mathrm{left}} \cdot \mathrm{diag}(\sigma_1, \sigma_2, \ldots, \sigma_K) \cdot \boldsymbol{\Phi}_{\mathrm{right}}^\top$, where $\boldsymbol{\Phi}_{\mathrm{left}} = (\phi_{ik}^{\mathrm{left}})_{ik} \in \mathbb{R}^{N \times K}$ and $\boldsymbol{\Phi}_{\mathrm{right}} = (\phi_{ik}^{\mathrm{right}})_{ik} \in \mathbb{R}^{N \times K}$ are the left and right singular vectors of $\tilde{\mathbf{R}}$ associated with the $K$ largest singular values ($\sigma_1, \ldots \sigma_K$) in magnitude, respectively. Then, we compute $\boldsymbol{u}_i$ and $\boldsymbol{v}_j$ by $u_{ik} = \sigma_k^\alpha \phi_{ik}^{\mathrm{left}}$, $v_{ik} = \sigma_k^{1-\alpha} \phi_{ik}^{\mathrm{right}}$ with $\alpha = 0.5$ following the previous studies [30, 32].

The analytical computation of $P_{\mathrm{d}}(j \mid i)$ is expensive because it scales as $\mathcal{O}(TN^3)$, where $T$ is the window size [32]. Alternatively, one can simulate random walks to estimate $P_{\mathrm{d}}(j \mid i)$.

Table 2: List of empirical graphs and structural properties. Variables $N$ and $M$ indicate the number of nodes and edges, respectively.

| Network | $N$ | $M$ | Assortativity | Max. degree | Clustering coef. | Ref. |
|---|---|---|---|---|---|---|
| Airport | 3,188 | 18,834 | -0.02 | 911 | 0.493 | [33] |
| Protein-Protein | 3,852 | 37,840 | -0.10 | 593 | 0.148 | [34] |
| Wikipedia vote | 7,066 | 100,736 | -0.08 | 1065 | 0.142 | [35] |
| Coauthorship (HepTh) | 8,638 | 24,827 | 0.24 | 65 | 0.482 | [36] |
| Citation (DBLP) | 12,494 | 49,594 | -0.05 | 713 | 0.118 | [37] |
| Coauthorship (AstroPh) | 17,903 | 197,031 | 0.20 | 504 | 0.633 | [36] |

Yet, both approaches require $\mathcal{O}(N^2)$ space complexity. Here, we reduce the time and space complexity by *the block approximation* that approximates the given graph by the dcSBM with $\hat{B}$ groups (we set $\hat{B} = 1,000$) and then computes an approximated $P_{\mathrm{d}}(j \mid i)$. The block approximation reduces the time and space complexity to $\mathcal{O}((N+M)\hat{B} + T\hat{B}^3)$ and $\mathcal{O}(NB\hat{B})$ for a graph with $N$ nodes and $M$ edges, respectively, with a high accuracy (the average Pearson correlation of 0.85 for the graphs in Table 2). See Supplementary Information for the block approximation.

## 4 Results

We test `residual2vec` using link prediction and community detection benchmarks [4, 22, 38–40]. We use the soft configuration model [31] as the null graph for `residual2vec`, denoted by `r2v-config`, which yields a degree-debiased embedding. The soft configuration model allows self-loops and multi-edges—which are not present in the graphs used in the benchmarks—and thus is not perfectly compatible with the benchmark graphs. Nevertheless, because the multi-edges and self-loops are rare in the case of sparse graphs, the soft configuration model has been widely used for sparse graphs without multi-edges and self-loops [23, 31, 39].

As baselines, we use (i) three random-walk-based methods, `node2vec` [4], `DeepWalk` [3], and `FairWalk` [16], (ii) two matrix-factorization-based methods, `Glove` [41] and Laplacian eigenmap (`LEM`) [42], and (iii) the graph convolutional network (`GCN`) [43], the graph attention networks (`GAT`) [44], and `GraphSAGE` [45]. For all random-walk-based methods, we run 10 walkers per node for 80 steps and set $T = 10$ and training iterations to 5. We set the parameters of `node2vec` by $p = 1$ and $q \in \{0.5, 1, 2\}$. For `Glove`, we input the sentences generated by random walks. We use two-layer `GCN`, `GraphSAGE`, and `GAT` implemented in StellarGraph package [46] with the parameter sets (e.g., the number of layers and activation function) used in [43–45]. Because node features are not available in the benchmarks, we alternatively use degree and eigenvectors for the $\hat{K}$ smallest eigenvalues of the normalized Laplacian matrix because they are useful for link prediction and clustering [47, 48]. We set $\hat{K}$ to dimension $K$ (i.e., $\hat{K} = K$). Increasing $\hat{K}$ does not improve performance much (Supplementary Information).

**Link prediction** Link prediction task is to find missing edges based on graph structure, a basic task for various applications such as recommending friends and products [4, 40, 49]. The link prediction task consists of the following three steps. First, given a graph, a fraction ($\rho = 0.5$) of edges are randomly removed. Second, the edge-removed graph is embedded using a graph embedding method. Third, the removed edges are predicted based on a likelihood score calculated based on the generated embedding. In the edge removal process, we keep edges in a minimum spanning tree of the graph to ensure that the graph is a connected component [4, 40]. This is because predicting edges between disconnected graphs is an ill-defined task because each disconnected component has no relation to the other.

We leverage both embedding $\boldsymbol{u}_i$ and baseline probability $P_0(j|i)$ to predict missing edges. Specifically, we calculate the prediction score by $\boldsymbol{u}_i^\top \boldsymbol{u}_j + z_i + z_j$, where we set $z_j = \ln P_0(j|i)$ for residual2vec because $\ln P_0(j|i)$ has the same unit as $\boldsymbol{u}_i^\top \boldsymbol{u}_j$ (Supplementary Information). Glove has a bias term that is equivalent to $z_i$. Therefore, we set $z_i$ to the bias term for Glove. Other methods do not have the parameter that corresponds to $z_i$ and thus we set $z_i = 0$. We measure the performance by the area under the curve of the receiver operating characteristics (AUC-ROC) for the prediction scores, with the removed edges and the same

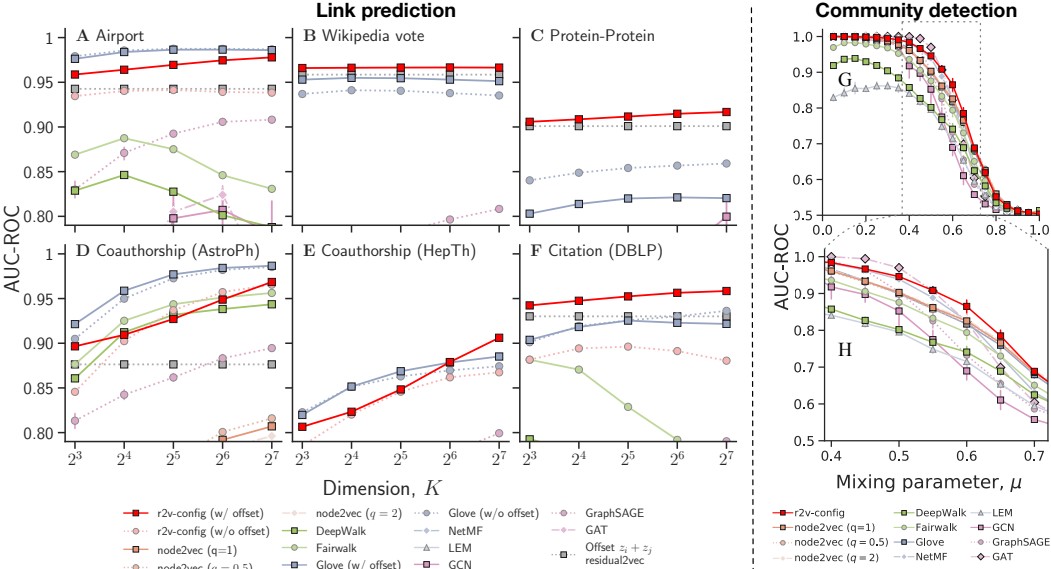

Figure 3: Performance for link prediction and community detection. We average AUC-ROC values over 30 runs with different random seeds and compute the 90% confidence interval by a bootstrapping with $10^4$ resamples. For link prediction, some underperforming models are not shown. See Supplementary Information for the full results. `r2v-config` consistently performs the best or nearly the best for all graphs in both benchmarks, as indicated by the higher AUC-ROC values.

number of randomly sampled non-existent edges being the positive and negative classes, respectively. We perform the benchmark for the graphs in Table 2.

`r2v-config` performs the best or nearly the best for all graphs (Figs. 3A–F). It consistently outperforms other random walk-based methods in all cases despite the fact that `node2vec` and `r2v-config` train the same model. The two methods have two key differences. First, `r2v-config` uses baseline $P_0(\ell \mid i) = P_d(\ell)$, whereas `node2vec` uses $P_0(\ell \mid i) \propto P_d(\ell)^{3/4}$ that does not exactly fit to the degree bias. Second, `r2v-config` optimizes the model based on a matrix factorization, which often yields a better embedding than the stochastic gradient descent algorithm used in `node2vec` [30, 32]. The performance of `residual2vec` is substantially improved when incorporating offset $z_i$, which itself is a strong predictor as indicated by the high AUC-ROC.

**Community detection** We use the Lancichinetti–Fortunato–Radicchi (LFR) community detection benchmark [38]. The LFR benchmark generates graphs having groups of densely connected nodes (i.e., communities) with a power-law degree distribution with a prescribed exponent $\tau$. We set $\tau = 3$ to generate the degree heterogeneous graphs. See Supplementary Information for the case of degree homogeneous graphs. In the LFR benchmark, each node has, on average, a specified fraction $\mu$ of neighbors in different communities. We generate graphs of $N = 1,000$ nodes with the parameters used in Ref. [38] and embed the graphs to $K = 64$ dimensional space. We evaluate the performance by randomly sampling $10,000$ node pairs and calculate the AUC-ROC for their cosine similarities, with nodes in the same and different communities being the positive and negative classes, respectively. A large AUC value indicates that nodes in the same community tend to have a higher similarity than those in different communities.

As $\mu$ increases from zero, the AUC for all methods decreases because nodes have more neighbors in different communities. `DeepWalk` and `LEM` have a small AUC value even at $\mu = 0.05$. `r2v-config` consistently achieves the highest or the second-highest AUC.

**Case study** Can debiasing reveal the salient structure of graphs more clearly? We construct a journal citation graph using citation data between 1900 and 2019 indexed in the Web of

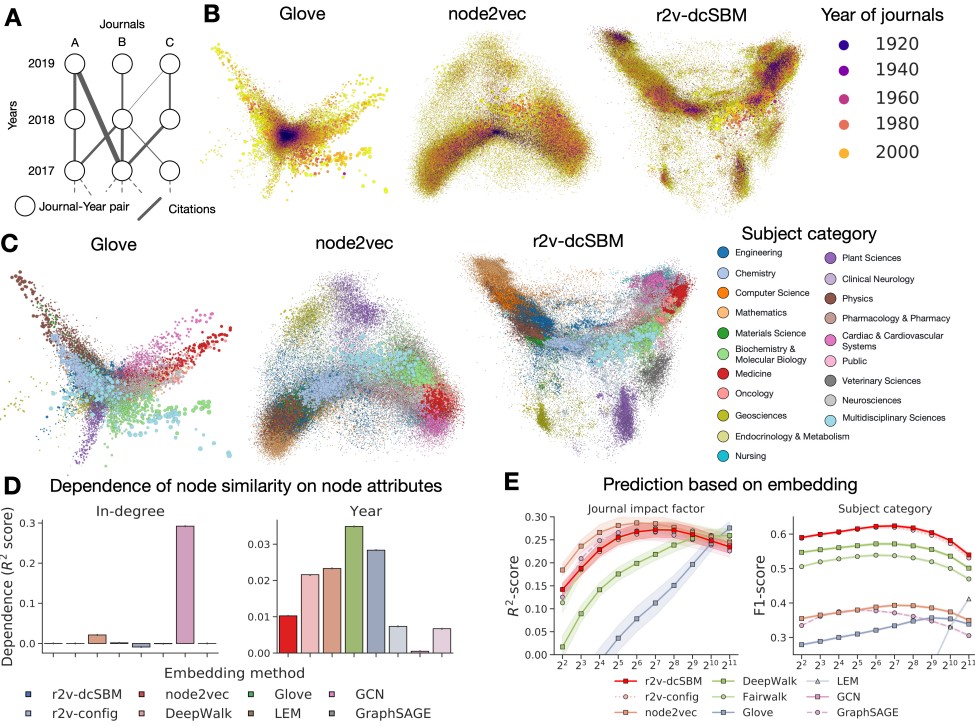

Figure 4: Embedding of the WOS journal citation graph. (**A**) Each node represents a pair $(j, t)$ of journal $j$ and year $t$, and each edge is weighted by citations. (**B**, **C**) A 2d projection of 128-dimensional embedding by the LDA. (**D**) `r2v-dcSBM` produces the embedding that is less dependent on degree and time. (**E**) By using the k-nearest neighbor algorithm, the embedding by `r2v-dcSBM` best predicts the impact factor and subject category.

Science (WoS) (Fig. 4A). Each node represents a pair $(i, t)$ of journal $i$ and year $t$. Each undirected edge between $(i, t)$ and $(j, t')$ is weighted by the number of citations between journal $i$ in year $t$ and journal $j$ in year $t'$. The graph consists of $242,789$ nodes and $254,793,567$ undirected and weighted edges. Because the graph has a high average degree (i.e., $1,049$), some algorithms are computationally demanding. For this reason, we omit `node2vec` with $q = 0.5$ and $q = 2$, `NetMF`, and `GAT` due to memory shortage (1Tb of RAM). Furthermore, for `GCN`, we set $\hat{K} = 32$, which still took more than 18 hours. We also perform the whole analysis for the directed graph to respect the directionality of citations. Although all methods perform worse in predicting impact factor and subject category, we find qualitatively the same results. See Supplementary Information for the results for the directed graph.

Here, in addition to the degree bias, there are also temporal biases, e.g., there has been an exponential growth in publications, older papers had more time to accumulate citations, and papers tend to cite those published in prior few years [50]. To remove both biases, we use `residual2vec` with the dcSBM (denoted by `r2v-dcSBM`), where we group journals by year to randomize edges while preserving the number of citations within and between years. We generate $K = 128$ dimensional embeddings with $T = 10$.

Figures 4B and C show the 2d projection by the Linear Discriminant Analysis (LDA) with journals' subject categories as the class labels. `Glove` and `node2vec` capture the temporal structure prominently, placing many old issues at the center of the embeddings. By contrast, `r2v-dcSBM` spreads out the old issues on the embedding. To quantify the effect of temporal bias, we randomly sample $50,000$ node pairs $(i, j)$ and then fitting a linear regression model $y_{ij} = w_0 + w_1(x_i + x_j) + w_2|x_i - x_j| + w_3 x_i x_j$ that predicts cosine similarity $y_{ij}$ for node pair $(i, j)$ with attributes $x_i$ and $x_j$, where $x_i$ is either the degree or the year of node $i$. We perform 5-cross validations and compute $R^2$-score (Fig. 4D). A smaller $R^2$-score indicates

that node similarity is less dependent on node attributes and thus less biased. `LEM` has the smallest $R^2$-score for both degree and year. `r2v-dcSBM` has a smaller $R^2$-score than `r2v-config` for year, respectively, suggesting that `r2v-dcSBM` successfully negates the biases due to time.

Is debiasing useful to capture the more relevant structure of graphs? We use embedding vectors to predict journal's impact factor (IF) and subject category. By employing the $k$-nearest neighbor algorithm, we carry out 5-cross validations and measure the prediction performance by $R^2$-score and the micro-F1 score. To ensure that the train and test sets do not have the same journals in the cross-validation, we split the set of journals $i$ into the train and test sets instead of splitting the set of nodes $(i,t)$. No single method best predicts both impact and subject categories. Yet, `r2v-config` and `r2v-dcSBM` consistently achieve the strongest or nearly the strongest prediction power for all $k$ we tested (Fig. 4E). This result demonstrates that debiasing embedding can reveal the salient structure of graphs that is overshadowed by other systematic biases.

## 5 Discussion

In this paper, starting from the insight that `word2vec` with SGNS has a built-in debiasing feature that cancels out the bias due to the degree of nodes, we generalize this debiasing feature further, proposing a method that can selectively remove any structural biases that are modeled by a null random graph. By exposing the bias and explicitly modeling it, we provide a new way to integrate prior knowledge about graphs into graph embedding, and a unifying framework to understand structural graph embedding methods. Under our residual2vec framework, other structural graph embedding methods that use random walks can be understood as special cases with different choices of null models. Through empirical evaluations, we demonstrate that debiasing improves link prediction and community detection performances, and better reveals the characteristics of nodes, as exemplified in the embedding of the WoS journal citation graph.

Our method is highly flexible because any random graph model can be used. Although we focus on two biases arising from degree and group structure in a graph, one can remove other biases such as the degree-degree correlation, clustering, and bipartitivity by considering appropriate null graphs. Beyond these statistical biases, there have been growing concerns about social bias (e.g., gender stereotype) as well as surveillance and privacy in AI applications, which prompted the study of gender and frequency biases in word embedding [51–54]. The flexibility and power of our selective and explicit debiasing approach may also be useful to address such biases that do not originate from common graph structures.

There are several limitations in `residual2vec`. We assume that random walks have a stationary distribution, which may not be the case for directed graphs. One can ensure the stationarity in random walks by randomly teleporting walkers [55]. Second, it is not yet clear to what extent debiasing affects downstream tasks (e.g., by losing information about the original graph). Nevertheless, we believe that the ability to understand and control systematic biases is critical to model graphs through the prism of embedding.

## Broader Impact

There has been an ever-increasing concern on inappropriate social stereotyping and the leak of sensitive information in word and graph embeddings [51, 56]. Although we have not studied social biases in this paper, given the wide usage of graph embedding methods to model social data, our approach may lead to methods and studies that expose and mitigate social biases that manifest as structural properties in graph datasets. Our general idea and approach may also be applied to modeling natural language and may contribute to the study of biases in language models. At the same time, by improving the accuracy of graph embedding, our method may also have negative impacts such as privacy attacks and exploitation of personal data (surveillance capitalism) [56, 57]. Nevertheless, we believe that our approach contributes to the effort to create transparent and accountable machine learning methods, especially because our method enables us to explicitly model what is structurally expected.

## Disclosure of Funding

The authors acknowledge support from the Air Force Office of Scientific Research under award number FA9550-19-1-0391.

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
