# Supplementary Information: Residual2Vec: Debiasing graph embedding with random graphs

Kojaku et al.

# Contents

# 1 Supplementary information for derivation

## 1.1 $P_{\mathrm{d}}(j \mid i)$ for the given graph

We compute conditional probability $P(j \mid i)$ for the given and random graphs to construct matrix $\mathbf{R}$ to factorize. While many methods simulate random walks, we can analytically compute $P(j \mid i)$ by exploiting the properties of the random walks. Denoted by $(x_1, x_2, x_3, \ldots)$ the sentence generated by a random walk in an undirected graph. For center-context node pair $(i, j)$, $P(j \mid i)$ is given by

$$P_{\mathrm{d}}(j \mid i) = \frac{1}{2T} \sum_{\tau=1}^{T} P_{\mathrm{rw}}(x_{t-\tau} = j \mid x_t = i) + \frac{1}{2T} \sum_{\tau=1}^{T} P_{\mathrm{rw}}(x_{t+\tau} = j \mid x_t = i), \qquad (1)$$

where $P_{\mathrm{rw}}(x_{t'} = j \mid x_t = i)$ is the probability that the walker visits $j$ at time $t'$ given that it visits $i$ at time $t$. The first and second terms of the right-hand side of Eq. (1) represent the probabilities of visiting $j$ *before* and *after* visiting $i$, respectively. Both terms are equal for an undirected and connected graph thanks to the following two key properties of the random walks. First, the random walk has a time-invariant stationary distribution, i.e., $P_{\mathrm{rw}}(x_{t-\tau} = j) = P_{\mathrm{rw}}(x_t = j) = P_{\mathrm{rw}}(j)$, where $P_{\mathrm{rw}}(j)$ is the stationary distribution at node $j$ [10]. Second, the random walks satisfy the detailed balanced condition [10], i.e., $P_{\mathrm{rw}}(i)P_{\mathrm{rw}}(x_t = j \mid x_{t-\tau} = i) = P_{\mathrm{rw}}(j)P_{\mathrm{rw}}(x_t = i \mid x_{t-\tau} = j)$. With these properties as well as the chain rule of probability, the first term can be rewritten as the second term:

$$P_{\mathrm{rw}}(x_{t-\tau} = j \mid x_t = i) = \frac{P_{\mathrm{rw}}(j)P_{\mathrm{rw}}(x_t = i \mid x_{t-\tau} = j)}{P_{\mathrm{rw}}(i)} = P_{\mathrm{rw}}(x_t = j \mid x_{t-\tau} = i). \qquad (2)$$

Returning to Eq. (1), conditional probability $P(j \mid i)$ can be computed analytically by

$$P_{\mathrm{d}}(j \mid i) = \frac{1}{T} \sum_{\tau=1}^{T} P_{\mathrm{rw}}(x_{t+\tau} = j \mid x_t = i) = \left( \frac{1}{T} \sum_{t=1}^{T} \mathbf{P}^t \right)_{ij}, \qquad (3)$$

where $\mathbf{P} = (P_{ij})$ is the *transition matrix* of random walks in the graph, with entry $P_{ij}$ indicating the probability of moving from $i$ to $j$ by one step.

## 1.2 $P_0(j \mid i)$ for the dcSBM

In the dcSBM, each edge between nodes $i$ and $j$ appears independently and has a weight $w_{ij}$ following a Poisson distribution with mean

$$\langle \tilde{w}_{ij} \rangle = \lambda_{g_i, g_j} \theta_i \theta_j, \qquad (4)$$

where $\langle \cdot \rangle$ is the expectation over the Poisson distribution, and $\{g_i\}_i$, $\{\lambda_{g,g'}\}_{g,g'}$, and $\{\theta_i\}_i$ are the parameters of the dcSBM. Parameter $g_i$ indicates the group to which node $i$ belongs, and $\lambda_{g,g'}$ represents the number of edges between groups $g$ and $g'$. Parameter $\theta_i$ is a propensity of node $i$ normalized such that

$$\sum_{i=1, g_i=g}^{N} \theta_i = 1, \ \forall 1 \leq g \leq B. \qquad (5)$$

Node propensity $\theta_i$ can be interpreted as a normalized degree of node $i$. In fact, the expected degree $\langle \tilde{d}_i \rangle$ of node $i$ is given by

$$\langle \tilde{d}_i \rangle = \left\langle \sum_{j=1}^N \tilde{w}_{ij} \right\rangle = \sum_{j=1}^N \lambda_{g_i,g_j} \theta_i \theta_j = \theta_i \sum_{g=1}^B \lambda_{g_i,g} \left( \sum_{j=1,g_j=g}^N \theta_j \right). \tag{6}$$

Because $\sum_{j=1,g_j=g}^N \theta_j = 1$ (Eq. (5)), we obtain

$$\theta_i = \frac{\langle \tilde{d}_i \rangle}{D_{g_i}}, \tag{7}$$

where $D_{g_i} = \sum_{g=1}^B \lambda_{g_i,g}$ is the number of edges emanating from nodes in group $g_i$, or equivalently, the sum of the degrees of nodes in group $g_i$.

For the graph with the expected edge weights, the transition probability from $i$ to $j$ is given by

$$P_{ij} = \frac{\langle \tilde{w}_{ij} \rangle}{\langle \tilde{d}_i \rangle} = \frac{\lambda_{g_i,g_j} \theta_i \theta_j}{\theta_i D_{g_i}} = \frac{\lambda_{g_i,g_j}}{D_{g_i}} \theta_j = P_{g_i,g_j}^{\mathrm{SBM}} \theta_j, \tag{8}$$

where we define

$$P_{g,g'}^{\mathrm{SBM}} := \lambda_{g,g'}/D_g, \tag{9}$$

by the fraction of edges to group $g'$ in $D_g$. Equation (8) makes clear that the transition of a walker from $i$ to $j$ in the dcSBM is decomposed into two random processes. The first is that a walker in group $g$ moves to group $g'$ with probability $P_{g,g'}^{\mathrm{SBM}}$. The second is that, in group $g'$, the walker lands on node $j$ with probability $\theta_j$.

With this interpretation in mind, let us consider multi-step random walks. As a concrete example, let us consider the transition of a walker with two steps. The probability of moving from $i$ to $j$ by $t = 2$ steps via node $x$ is given by $P_{i,x}P_{x,j} = P_{g_i,g_x}^{\mathrm{SBM}} P_{g_x,g_j}^{\mathrm{SBM}} \theta_x \theta_j$. Summing over all nodes $x$ yields the probability of transiting from $i$ to $j$ with two steps:

$$\sum_{x=1}^N P_{i,x}P_{x,j} = \theta_j \sum_{x=1}^N P_{g_i,g_x}^{\mathrm{SBM}} P_{g_x,g_j}^{\mathrm{SBM}} \theta_x = \theta_j \sum_{g=1}^B P_{g_i,g}^{\mathrm{SBM}} P_{g,g_j}^{\mathrm{SBM}} \left( \sum_{x=1,g_x=g}^N \theta_x \right)$$

$$= \theta_j \sum_{g=1}^B P_{g_i,g}^{\mathrm{SBM}} P_{g,g_j}^{\mathrm{SBM}}$$

$$= \left( \mathbf{P}_{\mathrm{SBM}}^2 \right)_{g_i,g_j} \theta_j, \tag{10}$$

where $\mathbf{P}_{\mathrm{SBM}} = (P_{ij}^{\mathrm{SBM}})_{ij}$. Notice that the transition probability for the two-step random walk takes the same form with that for the one-step random walk (i.e., Eq. (8)), which is also true for the $t$-step random walk, i.e.,

$$\left( \mathbf{P}^t \right)_{ij} = \left( \mathbf{P}_{\mathrm{SBM}}^t \right)_{g_i,g_j} \theta_j. \tag{11}$$

Therefore, the $t$-step random walks from node $i$ to $j$ can be described as two independent events, i.e., an event that the random walker moves from the block of node $i$ to the block of node $j$ by $t$ steps (i.e., $(\mathbf{P}_{\mathrm{SBM}}^t)_{g_i,g_j}$), and another event that the walker chooses node $j$ to visit among the nodes in the block $g_j$ (i.e., $\theta_j$).

The maximum likelihood estimation of dcSBM fits the parameters such that the degree of each node is preserved on average [8] by setting node propensity $\theta_i = d_i/D_{g_i}$ [8], which leads

$$\left(\mathbf{P}^t\right)_{i,j} = \frac{d_j}{D_{g_j}} \left(\mathbf{P}^t_{\text{SBM}}\right)_{g_i,g_j}. \tag{12}$$

Substituting Eq. (12) into Eq. (3) yields:

$$P_0(j \mid i) = \frac{d_j}{TD_j} \left(\sum_{t=1}^{T} \mathbf{P}^t_{\text{SBM}}\right)_{g_i,g_j}. \tag{13}$$

## 1.3  $P_0(j \mid i)$ for the special cases of the dcSBM

Let us derive $P_0(j \mid i)$ for two common null models, the Erdős-Rényi random graph [3] and the configuration model [4]. The Erdős-Rényi random graph is equivalent to the dcSBM with $B = 1$ and $\theta_i = 1/N$ [8], which gives

$$P_0(j \mid i) = 1/N. \tag{14}$$

The configuration model is equivalent to the dcSBM with $B = 1$ block and $\theta_i = d_i/2M$ [8], which gives

$$P_0(j \mid i) = d_j/2M. \tag{15}$$

## 1.4  $P_{\text{d}}(i) = P_0(i)$

The probability that the walker moves from node $i$ to a neighbor $j$ is given by

$$P_{\text{d}}(j \mid i) = \frac{1}{d_i}. \tag{16}$$

The random walk process in an undirected graph is ergodic, which ensures a unique stationarity for $t \to \infty$, with the detailed balance condition [10]:

$$P_{\text{d}}(j \mid i)P(i) = P_{\text{d}}(i \mid j)P_{\text{d}}(j) = c, \tag{17}$$

where $c > 0$ is a positive constant. By substituting Eq. (16) into Eq. (17), we have

$$P_{\text{d}}(i) = \frac{c}{P_{\text{d}}(j \mid i)} = cd_i. \tag{18}$$

Because $\sum_i P_{\text{d}}(i) = 1$, we obtain

$$P_{\text{d}}(i) = \frac{d_i}{\sum_{\ell} d_{\ell}}. \tag{19}$$

Notice that the stationary distribution of the random walker is proportional to degree $d_i$ and irrespective of the structure of the graph. Now, because both the dcSBM and the original graph have the same degree sequence on expectation, probability $P_0(i)$ for the dcSBM is given by

$$P_0(i) = \frac{d_i}{\sum_{\ell} d_{\ell}}. \tag{20}$$

## 1.5 Embedding directed graphs

Directed graphs may break the assumptions for the stationarity of random walks as well as the detailed balanced condition. Both assumptions are needed to compute $P_{\text{rw}}(x_{t-\ell} = j \mid x_t = i)$, i.e., a probability that the walker visits $j$ *before* visiting $i$. To avoid calculating the probability, we adopt a sliding window covering only the context nodes $j$ that appear *after* the center node $i$. This simplifies Eq. (1) into

$$P_{\text{d}}(j \mid i) = \frac{1}{T} \sum_{\tau=1}^{T} P_{\text{rw}}(x_{t+\tau} = j \mid x_t = i), \tag{21}$$

which in turn leads to the same expression as Eq. (3) without the assumptions of the stationarity and detailed balanced condition. A downside is that if node $i$ does not have outgoing edges (dangling node), then $i$ does not have any context $j$. To ensure that all nodes have at least one context, we allow a random walker to move against the direction of edges with a small probability $\epsilon = 0.05$. If a random walker hits any dangling node, it moves against a randomly selected in-coming edge.

## 2 Implementation of `residual2vec`

---
**Algorithm 1** Pseudocode of `residual2vec`.

---
**Input:**
    $\mathbf{A}$: Adjacency matrix of a graph of $N$ nodes
    $K$: Embedding dimension
    $\boldsymbol{g}$: Block membership of nodes for a null model (optional)
**Output:**
    Node embedding $\mathbf{U}$, where each $i$th row indicates the embedding of node $i$.
  1: $\hat{\boldsymbol{g}} \leftarrow \text{FITTINGDCSBM}(\mathbf{A})$ *//Ref. [7]*
  2: $\hat{\mathbf{R}} \leftarrow \text{TRUNCATERESIDUALMATRIX}(\mathbf{A}, \boldsymbol{g}, \hat{\boldsymbol{g}})$
  3: $\boldsymbol{\lambda}, \mathbf{U} \leftarrow \text{RANDOMIZEDSVD}(\hat{\mathbf{R}}, K)$ *//Ref. [6]*
  4: $\mathbf{U} \leftarrow \mathbf{U} \cdot \text{diag}(\sqrt{\boldsymbol{\lambda}})$

---

`residual2vec` has three components, namely i.e., (i) the block approximation, (ii) truncation, and (iii) matrix factorization (see Algorithm 1). We walk through how to implement each component efficiently in the following.

## 2.1 Block approximation

**Approximating $P_{\text{d}}(j \mid i)$ with the dcSBM** Our implementation centered on *the block approximation*, which substantially reduces the computational burden. Remind that `residual2vec` factorizes the residual matrix (Eq. (13) in the main text):

$$R_{ij} = \ln P_{\text{d}}(j \mid i) - \ln P_0(j \mid i). \tag{22}$$

Computing the second term (i.e., $\ln P_0(j \mid i)$) is easy because it can be computed by taking the power of a $B \times B$ matrix (i.e., Eq. (13)), and $B \ll N$ in practice.

    Computing the other term $\ln P_{\text{d}}(j \mid i)$, however, is prohibitively difficult because it involves the power of $N \times N$ matrix, $\mathbf{P}$ (i.e., Eq. (3)), requiring the time and space

---

**Algorithm 2** TRUNCATERESIDUALMATRIX

---

**Input:**
    **A**: Adjacency matrix
    ***g***: Group membership for random graphs
    ***ĝ***: Group membership for the approximated graph

**Output:**
    **R̃**: Truncated residual matrix

1: $\hat{R}_{ij} \leftarrow 0$ //*Initialize*

    //*Calculate the tentative $\tilde{R}_{ij}$ by truncating* **L**
2: **for all** $i \in [1, N]$, $g \in [1, B]$ and $\hat{g} \in [1, \hat{B}]$ **do**
3:     **if** $h(i, g, \hat{g}) > 0$ **then**
4:         $\hat{R}_{ij} \leftarrow h(i, g, \hat{g})$ for all $g_j = g$ and $\hat{g}_j = \hat{g}$
5:     **end if**
6: **end for**

    //*Re-evaluate Eq. (27) for $S_{ij} > 0$*
7: **for all** the non-zero elements of **A do**
8:     $\hat{R}_{ij} \leftarrow \max(0, S_{ij} + h(i, g, \hat{g}))$
9: **end for**

---

complexities of $\mathcal{O}(N^3)$ and $\mathcal{O}(N^2)$, respectively. The block approximation remedies this problem by approximating the given graph with the dcSBM [8]. Specifically, we fit the dcSBM to the given graph using a maximum likelihood estimation [7]. We set the number $\hat{B}$ of groups to $\hat{B} = \min(N, 1000)$, where $N$ is the number of nodes. With Eq. (12), the transition matrix for the approximated graph, denoted by $\hat{\mathbf{P}} = (\hat{P}_{ij})$, is given by

$$\hat{P}_{ij} = \frac{d_j}{D_{g_j}} \hat{P}^{\text{SBM}}_{\hat{g}_i, \hat{g}_j}, \tag{23}$$

where $\hat{g}_i$ is the group to which the block approximation assigns node $i$. The $t$th power $\hat{\mathbf{P}}^t$ can be computed by

$$\left(\hat{\mathbf{P}}^t\right)_{ij} = \frac{d_j}{D_{g_j}} \left(\hat{\mathbf{P}}^t_{\text{SBM}}\right)_{\hat{g}_i, \hat{g}_j}. \tag{24}$$

It is the strength of this approximation that allows us to calculate the matrix power efficiently. Notice that $\hat{\mathbf{P}}^t$—which is the power of an $N \times N$ matrix—can be computed by $\hat{\mathbf{P}}^t_{\text{SBM}}$—which is the power of a smaller $\hat{B} \times \hat{B}$ matrix ($\hat{B} \leq N$). We take advantage of this property by using $\hat{\mathbf{P}}^t$ as the substitute of $\mathbf{P}^t$ for $t > 0$ in Eq. (3) to compute $\ln P_{\text{d}}(j \mid i)$, i.e.,

$$\ln P_{\text{d}}(j \mid i) = \ln \left[ \frac{1}{T} \left( \sum_{t=1}^{T} \mathbf{P}^t \right) \right] \simeq \ln \left[ \frac{1}{T} \left( \mathbf{P} + \mathbf{P} \sum_{t=1}^{T-1} \hat{\mathbf{P}}^t \right) \right]_{ij} \tag{25}$$

$$=: \ln \hat{P}_{\text{d}}(j \mid i).$$

By substituting Eq. (24) into Eq. (25), we have

$$\ln \hat{P}_{\text{d}}(j \mid i) = \ln \frac{1}{T} \left[ P_{ij} + \sum_{\ell=1}^{N} P_{i\ell} \left( \sum_{t=1}^{T-1} \hat{\mathbf{P}}^t_{\text{SBM}} \right)_{\hat{g}_\ell, \hat{g}_j} \frac{d_j}{D_{\hat{g}_j}} \right]. \tag{26}$$

**Accuracy of the block approximation** The accuracy of the block approximation hinges on window size $T$ and the community structure of the graph. The block approximation is exact when $T = 1$ and $T = \infty$. For $1 < T < \infty$, the accuracy depends on how well the dcSBM approximates the given graph. The dcSBM describes the graph structure in terms of the connectivities of groups (or communities) and the degree of nodes. Therefore, the block approximation is a good approximation if the given graph has a strong community structure that is well described by the dcSBM.

As a proof of concept, we tested the block approximation using the empirical graphs listed in Table 2 in the main text (Fig. 1). We measured the Pearson correlation coefficient, denoted by $\rho$, between the exact and approximated $\ln P_{\mathrm{d}}(j \mid i)$. The airport network—which has a strong community structure that well fits the dcSBM [11]—has the largest correlation, $\rho$. By contrast, the correlation is relatively small for the DBLP citation graph. Yet, for all the graphs, the correlation is relatively high on average ($\rho = 0.81$). Overall, the correlation tends to increases as window size $T$ increases, suggesting that $P_{\mathrm{d}}(j \mid i)$ is well approximated, particularly for relatively large window size.

## 2.2 Truncation

`residual2vec` truncates the residual matrix by

$$\tilde{R}_{ij} = \max(0, R_{ij}), \tag{27}$$

which costs time complexity of $\mathcal{O}(N^2)$ because $\mathbf{R} = (R_{ij})$ has $N^2$ elements. Here, we perform the truncation more efficiently by taking advantage of the block approximation.

The block approximation approximates the residual matrix by

$$
\begin{aligned}
R_{ij} &= \ln \frac{1}{T} \left[ P_{ij} + \sum_{\ell=1}^{N} P_{i\ell} \left( \sum_{t=1}^{T-1} \hat{\mathbf{P}}_{\mathrm{SBM}}^{t} \right)_{\hat{g}_{\ell}, \hat{g}_{j}} \frac{d_j}{D_{\hat{g}_j}} \right] - \ln \frac{d_j}{T D_{g_j}} \left( \sum_{t=1}^{T} \mathbf{P}_{\mathrm{SBM}}^{t} \right)_{g_i, g_j} \\
&= \ln \left[ P_{ij} + \sum_{\ell=1}^{N} P_{i\ell} \left( \sum_{t=1}^{T-1} \hat{\mathbf{P}}_{\mathrm{SBM}}^{t} \right)_{\hat{g}_{\ell}, \hat{g}_{j}} \frac{d_j}{D_{\hat{g}_j}} \right] - \ln \frac{d_j}{D_{g_j}} \left( \sum_{t=1}^{T} \mathbf{P}_{\mathrm{SBM}}^{t} \right)_{g_i, g_j}.
\end{aligned} \tag{28}
$$

We note that $g_i$ and $\hat{g}_i$ are different, i.e., $g_i$ indicates the group of node $i$ for random graphs, and $\hat{g}_i$ indicates the group for approximating the given graph by the block approximation. The key consequence of the block approximation is that $\mathbf{R}$ can be decomposed into a block matrix, $\mathbf{L}$, and a sparse matrix $\mathbf{S}$, i.e.,

$$\mathbf{R} = \mathbf{L} + \mathbf{S}. \tag{29}$$

By leveraging the nature of $\mathbf{L}$ and $\mathbf{S}$, the truncation can be performed efficiently in the following two steps. First, we calculate tentative $R_{ij}$ values by truncating $\mathbf{L}$. Each element of $\mathbf{L}$ takes one of a handful of $r$ ($r \ll N^2$) unique values. Therefore, $\mathbf{L}$—which consists of $N^2$ elements—can be truncated by truncating the $r$ values. Second, we go through each element $(i, j)$ for $S_{ij} > 0$ and re-evaluate Eq. (27), i.e., $\max(0, L_{ij} + S_{ij})$. Matrix $\mathbf{S}$ has $M$ non-zero elements, where $M$ is the number of edges. Therefore, $\mathbf{R}$ can be truncated by truncating $r + M$ values, which is a substantial reduction of computations compared to truncating all $N^2$ elements.

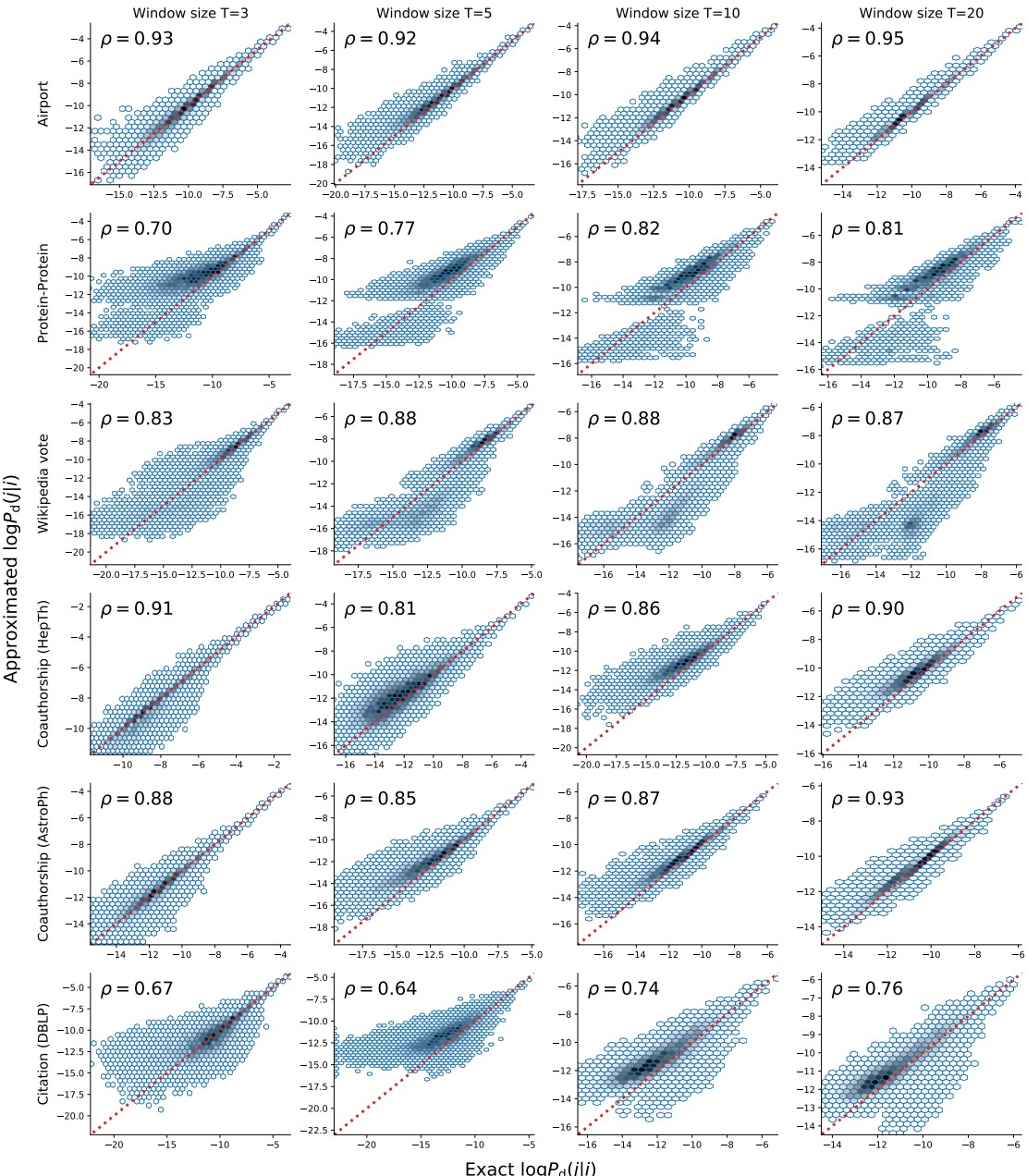

Figure 1: Approximated $P_{\mathrm{d}}(j \mid i)$ computed by the block approximation. The dashed diagonal line indicates the perfect match between the exact and approximated values. Variable $\rho$ in each panel indicates the Pearson correlation coefficient between the exact and approximated $\ln P_{\mathrm{d}}(j \mid i)$.

Specifically, $\mathbf{R}$ can be decomposed by

$$L_{ij} = \ln \sum_{\ell=1}^{N} P_{i\ell} \left( \sum_{t=1}^{T-1} \hat{\mathbf{P}}_{\text{SBM}}^t \right)_{\hat{g}_\ell, \hat{g}_j} - \ln \left( \sum_{t=1}^{T} \mathbf{P}_{\text{SBM}}^t \right)_{g_i, g_j} + \ln \frac{D_{g_j}}{D_{\hat{g}_j}}, \tag{30}$$

$$S_{ij} = \begin{cases} \ln \left( P_{ij} + \sum_{\ell=1}^{N} P_{i\ell} \left( \sum_{t=1}^{T-1} \hat{\mathbf{P}}_{\text{SBM}}^t \right)_{\hat{g}_\ell, \hat{g}_j} \right) \\ \quad - \ln \sum_{\ell=1}^{N} P_{i\ell} \left( \sum_{t=1}^{T-1} \hat{\mathbf{P}}_{\text{SBM}}^t \right)_{\hat{g}_\ell, \hat{g}_j} & (\text{if } P_{ij} > 0), \\ 0 & (\text{otherwise}). \end{cases} \tag{31}$$

Matrix $\mathbf{L}$ consists of at most $r = NB\hat{B}$ unique element values, i.e.,

$$L_{ij} \in \left\{ h(i, g, \hat{g}) \middle| i = 1, \ldots, N, g \in [1, B], \hat{g} \in [1, \hat{B}] \right\}, \tag{32}$$

where

$$h(i, g, \hat{g}) := \ln \sum_{\ell=1}^{N} P_{i\ell} \left( \sum_{t=1}^{T-1} \hat{\mathbf{P}}_{\text{SBM}}^t \right)_{\hat{g}_\ell, \hat{g}} - \ln \left( \sum_{t=1}^{T} \mathbf{P}_{\text{SBM}}^t \right)_{g_i, g} + \ln \frac{D_g}{D_{\hat{g}}}. \tag{33}$$

The pseudo code for the truncation is described in Table 2.

## 2.3   Matrix factorization

We factorize the truncated residual matrix, $\hat{\mathbf{R}}$, using the singular value decomposition (SVD). However, the SVD is practically infeasible for large graphs because its time complexity increases cubically with respect to $N$, i.e., $\mathcal{O}(N^3)$.

We circumvent this problem by leveraging the sparsity of $\hat{\mathbf{R}}$. The truncated residual matrix $\hat{\mathbf{R}}$ is sparse in our numerical simulations, e.g., at least 99% of the elements in $\mathbf{R}$ are zero for the six graphs. We take advantage of the sparsity of $\hat{\mathbf{R}}$ by using the randomized SVD (rSVD) [6]. The time and space complexities of the rSVD for computing $K$ leading eigenvectors are $\mathcal{O}((N+m)K)$ and $\mathcal{O}(NK)$, respectively, for an $N \times N$ sparse matrix with $m$ non-zero elements.

## 2.4   Computational complexity

The time and space complexities of each component in `residual2vec` are described in Table 1. We note that we used the rSVD for fitting the dcSBM instead of the SVD used in the original paper [7]. Taken together, the time complexity of `residual2vec` is $\mathcal{O}((N+M)\hat{B} + N\hat{B}^2 + TB^3 + M\hat{B} + T\hat{B}^3 + M + NB\hat{B} + \text{nnz}(N+\hat{\mathbf{R}})K + NK) = \mathcal{O}((N+M)\hat{B} + T\hat{B}^3)$, where we have assumed $B, K \leq \hat{B} \leq N$ and $\mathcal{O}(M) = \mathcal{O}(\text{nnz}(\hat{\mathbf{R}}))$. The space complexity is $\mathcal{O}(N\hat{B} + B^2 + N\hat{B} + B + \hat{B} + NB\hat{B} + NK) = \mathcal{O}(NB\hat{B})$.

# 3   Supplementary information for the experiments

## 3.1   Offset $z_i$ of `residual2vec` for link prediction

`residual2vec` decomposes node similarities into two components, i.e., embedding $u_i$ and baseline probability $P_0(j|i)$ in Eq. (10) in the main text. Baseline probability $P_0(j|i)$

Table 1: Time and space complexities of `residual2vec`.

| | Process description | Complexity Time | Space |
|---|---|---|---|
| Block approximation | Compute the $\hat{B}$ leading vectors of the adjacency matrix of the graph with $N$ nodes and $M$ edges using the rSVD [6]. | $\mathcal{O}((N+M)\hat{B}+\hat{B}^3)$ | $\mathcal{O}(N\hat{B})$ |
| | Perform the $K$-means clustering for $N$ nodes with $\hat{B}$ dimensional vectors [1]. | $\mathcal{O}(N\hat{B}^2)$ | $\mathcal{O}(N\hat{B})$ |
| Truncation | $\sum_{t=1}^{T}\mathbf{P}_{\text{SBM}}^t$ | $\mathcal{O}(TB^3)$ | $\mathcal{O}(B^2)$ |
| | $\sum_{\ell=1}^{N}P_{i\ell}\left(\sum_{t=1}^{T}\hat{\mathbf{P}}_{\text{SBM}}^t\right)_{g_\ell,g},\forall i\in[1,N]$ and $g\in[1,\hat{B}]$. | $\mathcal{O}(M\hat{B}+T\hat{B}^3)$ | $\mathcal{O}(N\hat{B})$ |
| | $D_g,\forall g\in[1,B]$ | $\mathcal{O}(M)$ | $\mathcal{O}(B)$ |
| | $D_{\hat{g}},\forall g\in[1,\hat{B}]$ | $\mathcal{O}(M)$ | $\mathcal{O}(\hat{B})$ |
| | Eq. (32) | $\mathcal{O}(NB\hat{B})$ | $\mathcal{O}(NB\hat{B})$ |
| Matrix factorization | Compute the $K$ leading vectors of $\hat{\mathbf{R}}$ using the rSVD [6]. | $\mathcal{O}(\text{nnz}((\hat{\mathbf{R}})+N)K)$ | $\mathcal{O}(NK)$ |
| | Scaling the dimensions by singular values | $\mathcal{O}(NK)$ | $\mathcal{O}(NK)$ |

accounts for the similarities attributed to a null model, and embedding similarity $u_i^\top u_j$ represents the "residual" from baseline probability $P_0(j|i)$. In the link prediction task, we aimed to leverage both baseline and residual similarities for prediction, by adding offset $z_j = \ln P_0(j|i)$ to the embedding similarity $u_i^\top u_j$. We added $\ln P_0$ instead of $P_0$ by noting that Eq. (10) in the main text can be rewritten as

$$P_{\text{r2v}}(j|i) = \frac{P_0(j|i)\exp(u_i^\top u_j)}{Z_i'} = \frac{\exp(u_i^\top u_j + \ln P_0(j|i))}{Z_i'}. \tag{34}$$

In other words, $\ln P_0(j|i)$ has the same unit as the embedding similarity $u_i^\top u_j$ in the model. Therefore, we adopted $\ln P_0(j|i)$ as the offset $z_i$.

## 3.2 Linear regression model for node similarities

In the Case study section, we quantified the strength of bias due to degree and time using a linear regression model $y_{ij} = f(x_i, x_j)$ that explains cosine similarity $y_{ij}$ for node pair $(i,j)$ by the degrees and the years of nodes $i$ and $j$, i.e., $x_i$ and $x_j$. A simple linear regression model is given by

$$\hat{y}_{ij} = w_0 + w_1 x_i + w_2 x_j, \tag{35}$$

Because the embedding similarity is symmetric ($y_{ij} = y_{ji}$), the modeled similarities should be also symmetric, i.e., $\hat{y}_{ij} = \hat{y}_{ji}$. By substituting Eq. (35) into $\hat{y}_{ij} = \hat{y}_{ji}$ yields $w_1 = w_2$, which leads

$$y_{ij} = w_0 + w_1(x_i + x_j). \tag{36}$$

This model explains the similarity by the sum of the node features, $x_i + x_j$. Instead of the summation, one can use the difference $|x_i - x_j|$ or product $x_i x_j$ for predicting the similarity $y_{ij}$. Here, we put them together into one linear regression model $\hat{y}_{ij} = w_0 + w_1(x_i + x_j) + w_2|x_i - x_j| + w_3 x_i x_j$. This composite model contains more variables and thus would explain the similarity $y_{ij}$ better than any of the model that only uses one variable.

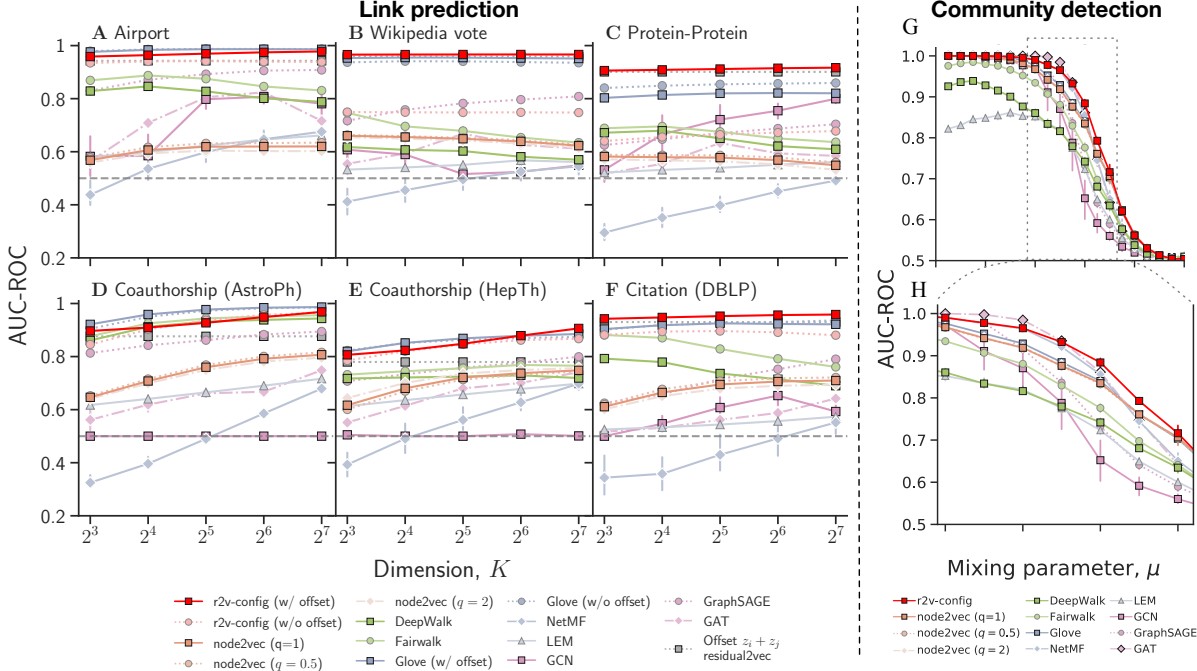

Figure 2: (A)—(F) The full results for the link prediction benchmark. (G–H) The results for the LFR model with homogeneous degree distributions. The power-law exponent for the degree distribution is set to $\gamma = 6$.

## 3.3 Full results for the link prediction benchmark

The full results for the link prediction benchmark are shown in Figs. 2A—F.

## 3.4 Community detection benchmark for graphs with homogeneous degree distributions

We performed the community detection benchmark with graphs having homogeneous degree distribution. Specifically, we generated the graphs using the LFR benchmark with the exponent of the degree distribution set to $\gamma = 6$. Other configurations for the benchmark are the same as those described in the main text. The results for the benchmark are shown in Figs. 2G and H.

## 3.5 GCN is not sensitive to the dimension of node features

In the benchmark, we use the eigenvectors associated with the smallest eigenvalues of the normalized Laplacian matrix as the input node features for GCN, GraphSAGE, and GAT. Although we used $\hat{K} = K$ eigenvectors, one can input more eigenvectors. Figure 3 shows the benchmark performance of the GCN and GraphSAGE with $\hat{K} = K$ or $\hat{K} = 2K$ eigenvectors as the input. Even if we input more eigenvectors ($\hat{K} = 2K$), the performance does not increase much or even gets worse. This may be because, for the link prediction and community detection tasks, $\hat{K} = K$ eigenvectors are sufficient. Furthermore, the more features we input, the more parameters the models have, making it difficult for the optimization algorithm to find a good parameter set.

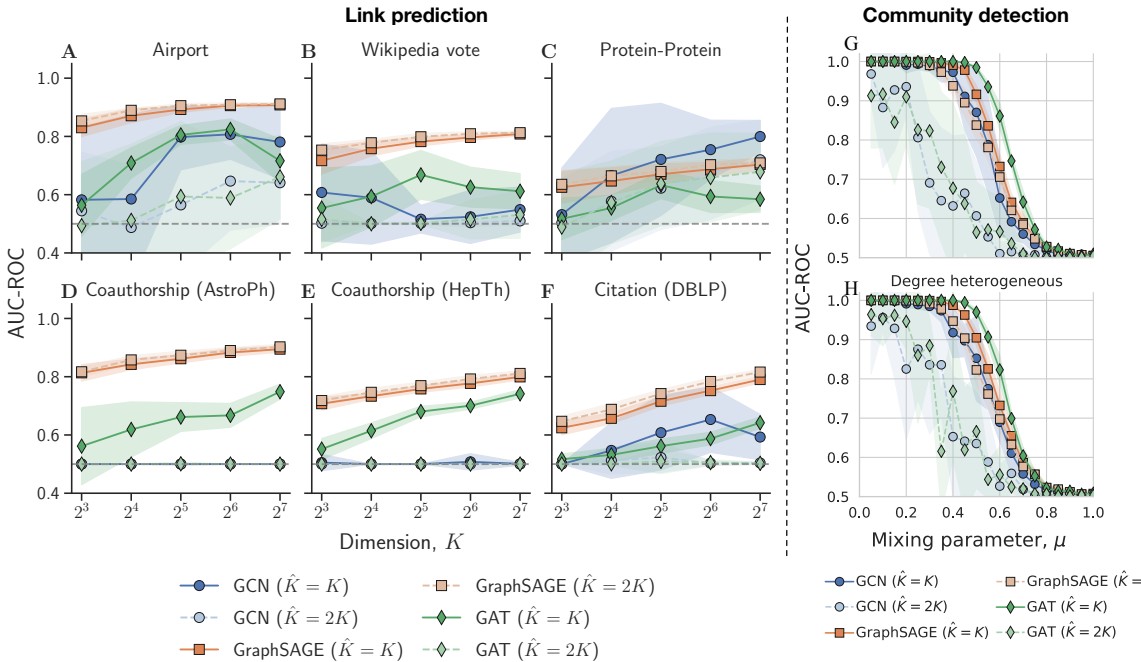

Figure 3: Performance for the link prediction and community detection benchmarks.

## 3.6 Results of directed citation graph of journals

Citations are inherently directional. However, we neglected the directionality and embedded the undirected citation graph of journals. This is because `residual2vec` and the baseline methods perform worse for the directed graphs in terms of the prediction of the impact and subject category of journals.

Figures 4 A and B show the 2D projection of the embedding generated by the Linear Discriminant Analysis, with the subject categories being the class labels. As is the case for the undirected graphs, `Glove` and `node2vec` strongly capture the temporal information. By contrast, `r2v-dcSBM` better delineates the subject categories more clearly than `Glove` and `node2vec`. In the embedding generated by `r2v-config`, node similarity—measured by the cosine similarity of the embedding vectors—is relatively independent of the degree and year compared to the embedding generated by `Glove`, `node2vec`, and `r2v-dcSBM` (Fig. 4C).

We predict the journals' impact factor and subject categories using the $k$-means algorithm with 5-cross validations. Although all methods predict worse compared to the embedding of the undirected graph, `r2v-config` best predicts the disciplines and impact of journals (Fig. 4D).

## 3.7 Web of Science citation graph

We construct a citation graph of journals using the citation data taken from the Web of Science (WoS) in 2020. The dataset contains bibliographic information including $1,547,459,602$ citations among $496,833,161$ papers published from $28,495$ journals in years between 1900 and 2019 in various research fields. We retrieve the subject category of each journal from the three journal collections, i.e., Science Citation Index Expanded (SCIE), Social Sciences Citation Index (SSCI), and Arts & Humanities Citation Index (AHCI). Each collection has a different category scheme. If a journal is indexed in mul-

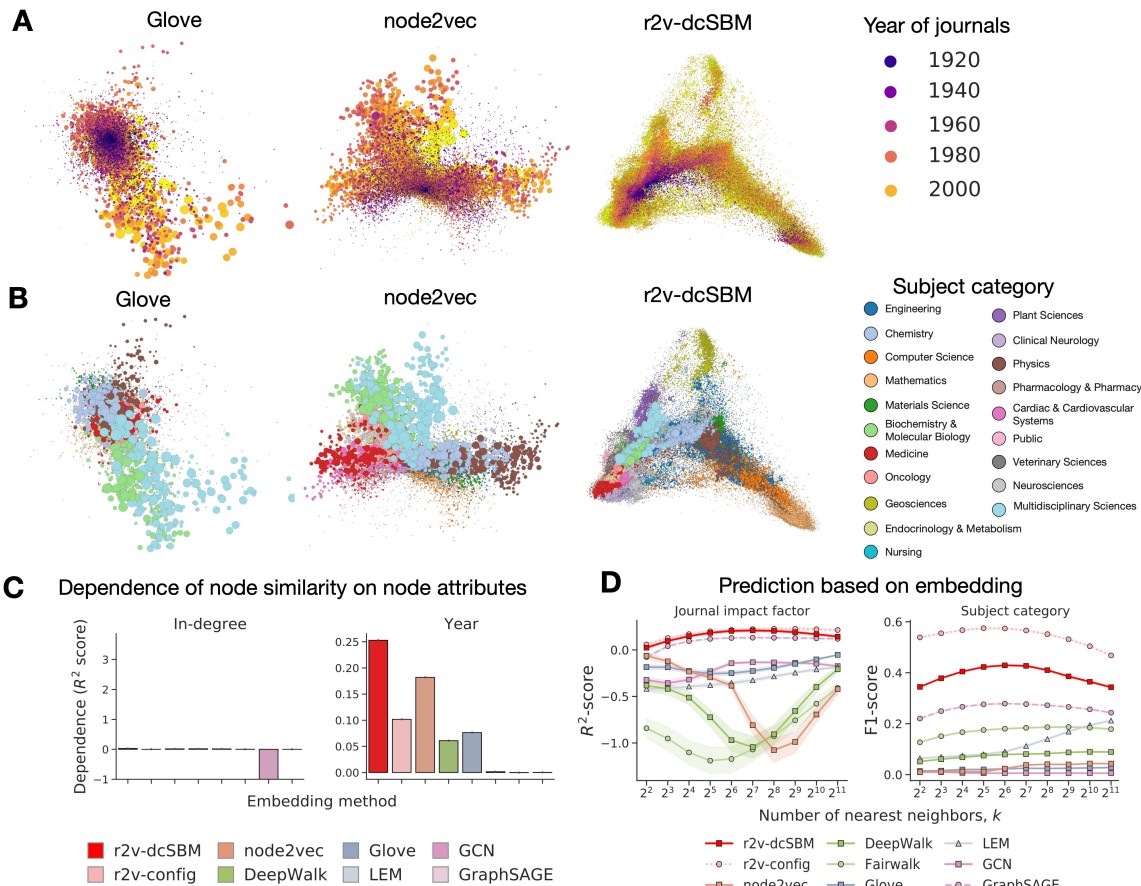

Figure 4: Embedding of the directed journal citation graph constructed from the WoS. (**A**, **B**) A 2d projection of 128-dimensional embedding by the Linear Discriminant Analysis. (**C**) Dependence of node similarity on nodes' degree and year. (**D**) By using the k-nearest neighbor algorithm, the embedding by `r2v-config` best predicts the impact factor and subject category.

tiple collections, we choose the largest collection and use its subject category for the journal. If the journal has multiple subjects within the chosen collection, we go through the table of the collection from the first to the last rows and use the one that first appears in the table.

## 3.8 Code

We implemented `DeepWalk` and `node2vec` using gensim package [12], with the same parameters used in the paper of node2vec [5]. We used `Glove` implemented in glove-python package [9]. We used `GCN` and `GraphSAGE` implemented in StellarGraph [2] and trained them using negative sampling.

## 3.9 Hardware

We ran experiments using a single machine with 64 Intel(R) Xeon(R) Gold 5218 CPU and 1Tb of RAM.