# OpenReview forum: "Residual2Vec: Debiasing graph embedding with random graphs"
_NeurIPS.cc/2021/Conference — NeurIPS 2021 Poster_

### Official Review · Reviewer_eKFw · 2021-07-16

**Rating:** 6
**Confidence:** 4

**Summary:**

This paper propose a prior probability for noise contrastive estimation (NCE) using degree-corrected stochastic block model (dcSBM), and propose a framework called residual2vec to learn the embedding of each node in the network, with a matrix factorization on the difference between random walk based empirical probabilities and the  dcSBM prior. The motivation arises from the implicit debiasing (of node degrees) property of negative sampling. Experiment results on link prediction and community detection are provided, as well as a case study on web of science citation network.

**Ethical Concerns:**

No Ethical Concerns

**Limitations And Societal Impact:**

Yes, they discuss them in section 5.

**Main Review:**

This work proposed a novel prior for NCE and matrix factorization solution of node embedding, based on the implicit debiasing (of node degrees) property of negative sampling from existing work, which is adequately cited. It is overall technically sound and clearly written, although some details and notations are not explicitly explained. The framework is potentially useful for practitioners to get better node embeddings with less dependency on node degrees, the web of science citation network can be a good dataset for other researchers to evaluate their algorithm if the authors can release them.

My comments and questions:

1. line 117, $\delta$ is not defined, it seems to be an indicator function. Please make sure to explain the notations as they appear.

2. lines 116-120, I can see from the supplementary that you are fitting a dcSBM first, please explicitly mention it, otherwise it confuses me how you get the group $g_i$

3. line 142, regarding $P_d(i)=P_0(i)$, it is not clear how you get it. Are you assuming what you learn from the dcSBM model is exactly the groundtruth, and even if so, how do you get it? Please provide a proof

4. line 248, why do you fit that particular linear regression model? The form is not commonly used.

5. For community detection experiment, why don't you use accuracy as the evaluation metrics? Also as you are using dcSBM as the null model in residual2vec, I would like to see how fitting a dcSBM itself performs in those experiments. If fitting dcSBM alone gives good community detection results, there is not much beneficial from the more complex residual2vec.

6. Will the web of science citation network be released for public validation? If not, I would suggest the authors to use other publicly available networks like arxiv or us patents from SNAP (http://snap.stanford.edu/data/index.html#citnets)

7. Figure 4D, it shows GCN has large dependence on degree but not year. Are you using negative sampling for GCN, if not, it does not seems to be a fair comparison to me.

Overall I think this paper is on the borderline. I like the idea of using dcSBM as null model and the explanations in sec 3.2, so I recommend borderline accept.

**Time Spent Reviewing:**

6

---

> ### Author Response · Authors · 2021-08-11
> **Reply to Reviewer eKFw**
>
> We appreciate the reviewer for careful reading of our paper. In the following, we go through the review point-by-point and address the reviewer's concerns one by one.
>
> ## Comment #1
>
> Thanks for spotting it. We have added text to explain the notation as follows.
>
> - (line 117)
>
>   *There are $D_{g}=\sum_{\ell=1}^N d_{\ell} \delta(g_\ell, g)$ edges emanating from group $g$, where $\delta$ is Kronecker delta.*
>
> Furthermore, we checked the manuscript and make sure that every notation is explained in the manuscript.
>
> ## Comment #2
>
> We employed two different dcSBMs for (i) generating random graphs and (ii) approximating a graph to reduce computation, which we should have delineated more clearly in our original text.
>
> (i) We assume that nodes have discrete labels (e.g., gender types) and use a dcSBM to remove biases attributed to the labels. If no such label is available, we consider that all nodes have the same label, which reduces the dcSBM to the configuration model. We use the discrete labels to determine the blocks of the dcSBM, where each block is a group of nodes having the same label.
>
> (ii) We also use another dcSBM for approximating the given graph to reduce the computation, which is described in Section 3.3. The results presented in the Supplementary Information show the accuracy of this approximation method and have nothing to do with the dcSBM as a graph null model.
>
> Nevertheless, we agree that we should have clarified this point better and thus have added the following text:
>
> * (line 116)
>
>   *Suppose that the nodes in the given graph have $B$ discrete labels (e.g., gender), and we want to remove the bias attributed to the labels.
>   If no such label is available, all nodes are considered to have the same label (i.e., $B=1$).
>   We fit the dcSBM with $B$ groups, where each group consists of the nodes with the same label.
>   With the dcSBM, we generate random graphs by randomizing the edges while keeping the number of edges within and between the groups on expectation (e.g., assortativity by gender types).
>   Then, we run random walks in the random graphs and calculate $P_{0}(j\vert i)$.
>   Here, we exploit the analytical properties of the dcSBM to calculate $P_{0}(j\vert i)$ without explicitly generating random graphs (Supplementary Information)*:
>
> ## Comment #3
>
> This equality always holds true for any assignment of nodes into blocks, regardless of whether the blocks are ground-truth or not. Let us first remind that $P_\text{d}(i)$ is the probability that a random walker visits node $i$ in the *given graph* (or equivalently the fraction of node $i$ in the walker's trajectory), and $P_\text{0}(i)$ is that in the *dcSBM*. Now, the key property of random walks is that $P_\text{d}(i)$ and $P_\text{0}(i)$ are determined by the degree of node $i$ regardless of the structure of the graphs [11]. Because the given graph and the dcSBM have the same degree for each node, we obtain $P_\text{d}(i) = P_\text{0}(i)$.
>
> To make this point more clear, we added the proof in the Supplementary Information and added text in the main text to refer to the proof.
>
> ## Comment #4
>
> We use this composite regression model due to the following reason.
> A simple linear regression model is
>
> $$
> \hat y_{ij} = w_0 + w_1 x_{i} + w_2 x_{j}, ...(1)
> $$
>
> where $\hat y_{ij}$ is the embedding similarity for nodes $i$ and $j$ predicted by the node features $x_i$ and $x_j$, where $x_i$ is the feature of node $i$, i.e., degree or year of a journal. Because the embedding similarity is symmetric ($y_{ij} = y_{ji}$), the predicted similarities should be also symmetric, i.e., $\hat y_{ij} = \hat y_{ji}$.
> By substituting (1) into $\hat y_{ij} = \hat y_{ji}$ yields $w_1 = w_2$, which leads
>
> $$
> y_{ij} = w_0 + w_1(x_{i} + x_j).
> $$
>
> This model predicts the similarity by the sum of the node features, $x_{i} + x_{j}$. Instead of the summation, one can use the difference $|x_i - x_j|$ or product $x_i x_j$ for predicting the similarity $y_{ij}$. Here, we put them together into one linear regression model $\hat y_{ij} = w_0 + w_1 (x_{i} + x_{j}) + w_2|x_i - x_j| + w_3x_i x_j$. This composite model contains more variables and thus would predict the similarity $y_{ij}$ better than any of the model that only uses one variable for prediction.
>
> We added text to clarify why we use this composite regression model in the Supplementary Information.
>
>
> ## Comment #5
>
> The dcSBM is by far the most commonly used method for community detection.
> However, we would like to note that, unlike the graph embedding, the dcSBM does not produce the vector-space representation of graphs. Because our primary purpose of the benchmarks is to compare the representations, we opted not to compare them with the dcSBM.
>
> We also would like to note that evaluating community detection problem is highly nontrivial [a1], and is sensitive to the choice of evaluation metrics and various settings. Because the focus of the paper is not on community detection, we deliberately used the simplest possible setting.
>
> We also would like to note that residual2vec uses the dcSBM to approximate graphs with $\hat B = 1000$ blocks. The approximation is performed only when there are more than $\hat B=1000$ nodes in the graph. In the LFR benchmark, the graph consists of $1000$ nodes. Therefore, the dcSBM is not used in residual2vec in the benchmark.
>
> [a1] Gates, Alexander J., Ian B. Wood, William P. Hetrick, and Yong Yeol Ahn. 2019. “Element-Centric Clustering Comparison Unifies Overlaps and Hierarchy.” Scientific Reports 9 (1): 1–13.
>
> ## Comment #6
>
> Unfortunately, although we fully acknowledge that sharing data is important to allow the verification of our results and to ensure reproducibility, we are not able to share the data publicly due to the data access contract. Due to the tight schedule of NeurIPS review rounds, we are afraid that we cannot redo the whole analysis with new datasets. However, we would like to note that, even without the exactly same dataset, there are other similar public bibliographic datasets such as Microsoft Academic Graph and DBLP, and it will be still possible to perform *replication studies*---which tends to provide greater value than *validation through reproduction*---using the public datasets.
>
> ## Comment #7
>
> Yes, we used the negative sampling framework to train GCN. We have added text to clarify this point in the Supplementary Information.
>
>
> In summary, we have updated our manuscript by clarifying our notations, our usage of the dcSBM, the justification of the use of the regression models, and the detailed parameter setting of the GCN. We appreciate the reviewer for careful reading of the paper and providing valuable input, which provided the opportunity to improve our manuscript.

---

### Official Review · Reviewer_RJD5 · 2021-07-17

**Rating:** 6
**Confidence:** 3

**Summary:**

The paper presents residual2vec, a random-walk based node embedding method that explicitly factors out the structural bias of a specified null graph model. The authors claim that, by factoring out the structural biases of a null graph model, the learned embeddings better capture the salient features of the graph. The method learns node embeddings in an efficient way using singular value decomposition. It is then compared with other random-walk based node embedding methods, including node2vec and DeepWalk. The experiments examine how the different methods perform in terms of link prediction, community detection, and the prediction of underlying node features from embeddings. In most cases, residual2vec appears to do as well or better than these baselines. They also train a linear regression model, for each of the methods, to predict the node cosine similarity using the node degrees. Residual2vec (and several of the baselines) achieve a low r-squared in this linear regression, indicating that the embeddings are not biased by node degree.

**Ethical Concerns:**

I have no ethical concerns about this paper

**Limitations And Societal Impact:**

I think the authors adequately address the limitations and societal impact

**Main Review:**

See the References section of the main paper for any citations used in this review.

## Originality:

Some parts of the paper are novel. The embeddings on their own are learned to solve the same problem as node2vec. Both the authors of word2vec and residual2vec treat the noise distribution as a free parameter. The novel contribution of this paper are as follow:
1.	It explicitly incorporates of the null graph transition probability in the conditional model (Equation 10).
2.	It shows that the model can be fit perfectly if the dot product of embeddings is equal to the residual pointwise mutual information (explained in the paper).
3.	The learning of the embeddings by singular value decomposition is an efficient solution.
4.	The experiments and results are novel.

I think the contribution of some other works should be stated more clearly. For instance, in section 2.2 Implicit debiasing by negative sampling, it should be made clear that equations 4, 5 and 6 come from [9].

I think the authors should clearly state the differences between residual2vec, and the skip-gram with negative sampling (NEG) by Mikolov et al. [6], where they describe the noise distribution as a free parameter of both NCE and NEG.

## Quality:

There are a few statements in the paper that I believe to be technically inaccurate (see detailed comments). I believe there is also an unstated assumption regarding the transition probability in the configuration model (also in the detailed comments).

I would be interested to see how the offset ln $d_i$ + ln $d_j$ performs on its own in the link prediction task, as this is a monotonic transformation of preferential attachment, which is often a highly predictive metric.

I am unconvinced that r2v-dcSBM achieves the best visual separation of classes in the 2d projection.

The prediction of journal impact factor and journal subject using KNN should ensure that test set nodes cannot have the same journal with a different year as one of its K nearest neighbours, and this should be made clear in the text.

## Clarity:

Most of the text is okay with regards to clarity, with the exception of section 2.2 Implicit debiasing by negative sampling. It took me a while to understand this section. I would like to see more of an introduction laying out the purpose of the derivation (to infer an unbiased model for $p(j|i)$), the logic of the derivation (showing that, for a certain choice of $f$, it has the same posterior probability as an estimator that we know to be unbiased), and the distinction between $Pm(x)$, $P(Yj=1|j)$ and $P(j|i)$.

One of the biggest issues with the paper is the figures displaying the results, particularly Figure 3. It is difficult to visually distinguish some of the series and their confidence intervals (due to overlap). A larger figure or a table might make the results clearer. Figure 3A-F also contain a series which is not in the legend (it is possible part of the legend has been cut off). Figure 3G-H has no explanation for the difference between G Degree homogeneous and H Degree heterogeneous in the main text nor the caption (I can guess, but clarification would be nice).

## Significance:

I think the novel elements of the paper are sufficiently significant for publication. I think it is possible that other researchers will use some of these ideas or build on them. Equation 9 in the paper helps to explain the widely used skip-gram negative sampling.

I have added some more detailed comments relating to specific parts of the paper below.

* Line 6-7: “Most notably, random walks are biased by the degree of each node, where, at each step, a node is sampled proportionally to its degree”

   Overall, a node is sampled proportionally to degree. At each step, I think neighbouring nodes are sampled uniformly (or in proportion to their share of edge weights of the present node).

* Line 53: “… SGNS word2vec models conditional probability…”

   Here, they define $P_{\text{w2v}}$ (a sigmoid) as the way in which SGNS word2vec models the conditional probability of the context given the centre word. This is problematic because using the softmax here has nothing to do with negative sampling. They correctly carry on pointing out that learning the log probability of the softmax over a large corpus is not feasible. This was originally pointed out by the word2vec paper, where they also introduced and compared several strategies, including hierarchical softmax, NCE, negative sampling, and subsampling of frequent words. I would ask the authors to refrain to using SGNS word2vec here, and use skip-gram instead to avoid confusion.

* Line 60: “Additionally, one samples k random context word…”

   This is slightly misleading as context words are not being sampled. Words are being randomly sampled as candidate context words.

* Line 70-71: “To see this, let us consider the unbiased variant of negative sampling, Noise Contrastive Estimation (NCE)”

   Here, the authors introduce NCE as a “the unbiased variant of negative sampling”. This is incorrect, as NCE precedes NEG, and Mikolov et al. (authors of word2vec) describe NEG as a simplification of NCE. I ask the authors to rephrase this and not call NCE a variant of NEG, in favour of something else. The authors also present insufficient evidence that NCE is unbiased. I ask that the authors either provide more evidence of this, or reduce their claim to asymptomatically unbiased (in the number of negative samples, $k$ [9]).

* Line 72: “where $f$ is a non-negative function of data $x$ in all data $\mathcal{X}$”

   Could the authors please clarify what they mean by ‘all data’ here?

* Section 3  Residual2vec graph embedding, Line 91-92

   The small paragraph at the start of this section mentions that the authors assume graphs are weighted, but it is not mentioned whether the graphs used in the evaluation are all weighted. The supplementary also mentions that the weights are Poisson distributed, which assumes that they only take discrete values, but this is not mentioned in the main text.

* Section 3.1 … Random graph models

   The dcSBM only preserves the expected degree of each node. Therefore, setting B=1 does not result in the configuration model, which exactly preserves the degree of each node.
Similarly, dcSBM only preserves the expected number of edges in the graph (according to the supplementary, the edge weights follow a Poisson distribution). Therefore, setting $B=1$ and $d_i =$ constant does not result in the Erdős-Rényi random graph that only preserves the number of edges.

* Table 1

   The baseline probability of $P_0(j|i)$ for the configuration model given in the table is an approximation for when the number of nodes is large, but this is never mentioned. It is not accurate for small graphs. Consider the configuration model defined by the degree sequence (a:1, b:1, c:2). For $T=1$, The nodes of degree 1 (a and b) cannot be linked within this model and thus $P_0(a|b) = 0$, and not $\frac{1}{4}$ as the table suggests.

If the authors can adequately address my concerns, I would consider increasing my score.

Edit: Based on the edits the authors have made in response to my review, I have edited my rating, increasing it from 4 to 6.


**Time Spent Reviewing:**

6

---

> ### Author Response · Authors · 2021-08-11
> **Reply to Reviewer RJD5**
>
> First of all, we appreciate the reviewer's careful reading of the paper and many constructive comments!
>
> # Originality
>
> We agree that we should have clarified our contribution to the previous studies more clearly. Therefore, we have added text as follows:
>
> * (line 105):
>
>     *We note that residual2vec with baseline $P_0(j \vert i)$ is equivalent to word2vec trained by negative sampling with noise distribution $P_0(j \vert i)$*.
>
> Furthermore, we have added citations before Eqs. (4)-(6) to indicate that they come from Ref. [9] (see the Clarity section in our response).
>
> # Quality
>
> ## Link prediction by offset
>
> This is a good point. We have added the prediction by the offset as a baseline. This prediction achieved the 96% of the AUC-ROC value of residual2vec on average, which is the third-highest following Glove (98%).
>
> ## The purpose of Figs. 4b and 4c.
>
> The visualization is intended to show the overall structure of the embedding to prepare readers to understand the following quantitative assessment in Fig. 4d and 4e, where the temporal biases and the separation of classes are measured using linear regression and a k-nearest neighbor algorithm. Nevertheless, we should emphasize this point, and thus have amended the text as follows:
>
> * (line 244)
>
>   *Glove and node2vec capture the temporal structure prominently, placing many old journals at the center of the embeddings. By contrast, r2v-dcSBM spreads out the old journals on the embedding. To quantify the effect of temporal bias, we randomly ...(continue)*
>
> ## Prediction of the impact and subject of journals
>
> Thank you for pointing at the important problem. We have corrected our procedure and added the text as follows:
>
> * (line 258)
>
>   *To ensure that the train and test sets do not have the same journal in the cross-validation, we split the set of journals $i$ into the train and test sets instead of splitting the set of nodes ($i$, $t$)*.
>
> By this change, the top-performers decreased their performance. For the prediction of the impact factor, node2vec achieved the highest $R^2$-score (i.e., 0.284) although it is comparable to that for the second-best performer, r2v-dcSBM (i.e., 0.277). For the prediction of subjects, r2v-dcSBM achieved the highest $F_1$-score (i.e., 0.622) followed by r2v-config (i.e., 0.619) and DeepWalk (i.e., 0.572). We would like to note that SOTA performance is not the primary contribution of our work and good performance is rather an excellent side effect of bias correction.
>
> # Clarity
>
> ## Clarity of Section 2.2
>
> Thank you for these valuable inputs, which helped us to improve our derivation. To make clear the derivation steps, we reorganized Section 2.2 into three parts as follows. First, we lay out the relationship between NEG and NCE, and the conclusion derived from the relationship. The second part is dedicated to explaining how NCE works. Third, we show how NCE is simplified into NEG and the impact of this simplification on embeddings.
>
> * (line 66) **Section 2.2 Implicit debiasing by negative sampling**
>
>   *Negative sampling efficiently produces a good representation [6]. An often overlooked fact is that negative sampling is a simplified version of Noise Contrastive Estimation (NCE) [8,9], and this simplification biases the model estimation. In the following, we show that this estimation bias gives rise to a debiasing feature of SGNS word2vec.*
>
>   **Noise contrastive estimation** *First, let us explain what NCE is. NCE is a generic estimator for probability model $P_m$ of the form:*
>
>   $$\text{Eq. (3)}$$
>
>   *where $f$ is a non-negative function of data $x$ in the set ${\cal X}$ of all possible values of $x$. word2vec (Eq. (1)) is a special case of $P_m$, where $f(x) = \exp(x)$ and $x = u_i ^\top v_{j}$. NCE estimates $P_m$ by solving the same task as negative sampling, i.e., classifying a positive and $k$ randomly sampled negative examples using logistic regression. However, NCE solves the task based on a Bayesian framework [8,9]. Specifically, as prior knowledge, we know that $1$ in $1+k$ pairs are taken from the given data, which can be expressed as prior probabilities [8,9]:*
>
>   $$
>   \text{Eq. (4)}
>   $$
>
>   *Assuming that the given data is generated from $P_m$, the positive example ($Y_j = 1$) and the negative examples ($Y_j = 0$) are sampled from $P_m$ and $p_0(j)$, respectively [8,9], i.e.,*
>
>   $$
>   \text{Eq. (5)}
>   $$
>
>   *Substituting Eqs. (4) and (5) into the Bayes rule yields the posterior probability for $Y_j$ given an example $j$ [8,9]:*
>
>   $$
>   \text{Eq. (6)}
>   $$
>
>   *which can be rewritten with a sigmoid function as*
>
>   $$
>   \text{Eq. (7)}
>   $$
>
>   *where $c = \ln k + \ln\sum_{ x' \in {\cal X}} f(x')$ is a constant. NCE uses Eq. (7) to classify data into positive and negative examples. The key feature of NCE is that it is an asymptomatically unbiased estimator of $P_m$ (Eq. (3)) whose bias goes to zero as the number of training examples goes to infinity [8].*
>
>   **Estimation bias of negative sampling** *In the original paper of word2vec [6], the authors simplified NCE into negative sampling by dropping $\ln p_0(j) + c$ in Eq. (7) because it reduced the computation and yielded a good word embedding. In the following, we show the impact of this simplification on the final embedding.*
>
>   *The model for negative sampling $P_{\text{NS}}$ (i.e., Eq.(2)) can be rewritten in the form of Eq. (7) as*
>
>   $$
>   \text{Eq. (8)}
>   $$
>
>   *Equation (8) makes clear the relationship between negative sampling and NCE: negative sampling is the NCE with $f(u_i ^\top v_{j}) = \exp\left( u_i ^\top v_{j} + \ln p_0(j) + c\right)$ and noise distribution $p_0$ [10].
>   Bearing in mind that NCE is the asymptomatically unbiased estimator of Eq. (3) and substituting $f(u_i ^\top v_{j})$ into Eq. (3), we show that SGNS word2vec is an asymptomatically unbiased estimator for probability model:*
>
>   $$
>   \text{Eq. (9)}
>   $$
>
>   *Equation (9) clarifies* ...(continue)
>
> ## Clarity of figure
> To make each method as distinguishable as possible, we have added new zoom-in panels to Fig. 3. We found that the legend was cut off, which we have amended. We also have added the following text to clarify the panel titles:
>
>   * (line 215)
>
>     *We set $\tau = 6$ or $\tau = 3$ to generate the degree homogeneous and heterogeneous graphs.*
>
> # Significance
>
> ### Line 6-7
>
> The reviewer is right. We have amended the text as follows:
>
> * (line 6)
>
>   *Most notably, random walks are biased by the degree of each node, sampling a node with probability proportionally to its degree.*
>
> ### Line 53
>
> We agree and have amended the text as follows:
>
> * (line 53)
>
>   *For a center-context word pair (i,j), word2vec models conditional probability*
>
> ### Line 60
>
> We have amended the text as follows:
>
> * (line 60)
>
>   *Additionally, one samples $k$ random word $\ell$ as candidate context words from a noise distribution $p_0(\ell)$, and then labels $(i,\ell)$ as $Y_{\ell} = 0$.*
>
> ### Line 70-71
>
> We agree. See the "Clarity of Section 2.2" section in our response.
>
> ### Line 72
>
> We agree and have amended the text as follows:
>
> * (line 72)
>
>   *where $f$ is a non-negative function of data $x$ in the set ${\cal X}$ of all possible values of $x$.*
>
> ### Section 3 Residual2vec graph embedding, Line 91-92
>
> We would like to note that unweighted graphs can be regarded as weighted graphs, where each edge has the same weight [a1-a3]. To clarify this point, we have added text as follows:
>
> - (line 92)
>
>   *We consider unweighted graphs as weighted graphs with all edge weights being one [12, 13]*.
>
> - (line 116)
>
>   *The dcSBM assumes that the weight of an edge follows a Poisson distribution and thus generates random graphs with discrete edge weights.*
>
> ### Section 3.1 Random graph models and Table 1
>
> First, we would like to note that there are two configuration models, namely *canonical* and *microcanonical* models [a4,a5], which we should have clarified in our text. The reviewer referred to the microcanonical model that preserves the degree exactly. On the other hand, we refer to the canonical model that preserves the degree on expectation. Similarly, there are the canonical and microcanonical Erdős-Rényi random graphs [a4].
>
> To clarify this point, we have amended the text as follows.
>
> * (line 112)
>
>   *The dcSBM can be mapped to many canonical random graph models. In fact, when $B=1$, the dcSBM is reduced to the configuration model that preserves the degree of each node on expectation [20]. Furthermore, by setting $B=1$ and $d_i = \text{constant}$, the dcSBM is reduced to the Erdős-Rényi random graph that only preserves the number of edges on expectation.*
>
> Furthermore, we have modified the header of Table 1 as follows.
>
> * (Table 1)
>
>   *Canonical random graph models*
>
> For the example of three nodes, a, b and c, we note that the configuration model allows self-loops. Therefore, if node a has a self-loop (note that a self-loop is double-counted [a5]), and b and c are connected by an edge, this graph is compatible with the degree sequence.
>
> In summary, we have updated the manuscript by (i) reorganizing Section 2.2 to improve the clarity of our derivation and to describe the relationships of NCE, NEG, and residual2vec correctly, (ii) adding a simple baseline to the benchmark, and (iii) correcting ambiguous wording and the errors. We deeply appreciate the reviewer's careful reading of our paper and many constructive comments, which greatly improved our manuscript.
>
> [a1] M. E. J. Newman. 2014. “Networks: An Introduction.” Oxford University, 163–86.
>
> [a2] Blondel, et al. 2008. “Fast Unfolding of Communities in Large Networks.” J. Stat. Mech.: Theory and Experiment 2008 (10): P10008.
>
> [a3] S. Fortunato. 2010. “Community Detection in Graphs.” Phys. Rep. 486 (3-5): 75–174.
>
> [a4] K. Anand, and G. Bianconi. 2009. “Entropy Measures for Networks: Toward an Information Theory of Complex Topologies.” Phys. Rev. E 80 (4): 45102.
>
> [a5] Fosdick et al. 2018. “Configuring Random Graph Models with Fixed Degree Sequences.” SIAM Review 60 (2): 315–55.

---

> > ### Comment · Reviewer_RJD5 · 2021-08-24
> > **Reply to Authors**
> >
> > Thank you for your reply, and the edits you have made in response to my feedback.
> >
> > Except for some additional comments below, I am happy that you have adequately addressed the concerns raised in my review. If the concerns below are also adequately addressed, I will increase my score.
> >
> > Section 3 Residual2vec graph embedding, Line 91-92
> >
> > I appreciate that unweighted graphs can be represented as weighted graphs with all weights set to 1. However, I think the text needs to compare the assumed graph type used to calculate the baseline probabilities, and the graph types used in the experiments. Specifically, whether they permit self-loops and integer-weighted/multi- edges. You ought to state that you assume that the baseline probability calculated for graphs with self-loops and integer-weighted/multi- edges is a good approximation for the graph types used in the experiments.
> >
> > Section 3.1 Random graph models and Table 1
> >
> > I am glad that the authors have clarified that they are preserving expected quantities in their graph models rather than exact quantities.
> >
> > One point I would like to add is that the term "canonical random graph model" does not seem to be standard in the literature. Instead, the term "canonical ensemble" is usually used, and understood in the context of a constraint (on an average quantity) in the graphs. If you prefer not to introduce the concept of canonical network ensembles, I think it is sufficient to explain that it is the expected quantities that are preserved. I would also suggest clarifications as to which version of the configuration and Erdős-Rényi models are assumed with regards to self-loops and multi-edges.
> >
> > In the case of preserved expected nodes degrees, I suggest referring to the model as the soft configuration model, with self-loops and multi-edges permitted.
> >
> > In the case of preserved expected number of edges, I suggest referring to the model as a multigraph version of the Erdős-Rényi random graph model as is done in [b1] (whilst still explaining that the expected number of edges is preserved).
> >
> >
> > For the example of three nodes a, b and c:
> >
> > There are many variants of the configuration model [b2]. If you allow self-loops, then b and c can be connected. Provided the column heading is changed to "soft configuration model", and the text makes it clear that self-loops and multi-edges are assumed in the calculation, I am happy with the baseline probability listed in Table 1.
> >
> >
> > [b1] Karrer, B. & Newman, M. E. J. Stochastic blockmodels and community structure in networks. Physical Review E 83, 016107. issn: 1539-3755. http://link.aps.org/doi/10.1103/PhysRevE.83.016107 (2011)
> >
> > [b2] Fosdick, B, Larremore, D, Nishimura, J & Ugander, J. Configuring Random Graph Models with Fixed Degree Sequences. SIAM Review 60, 315–355 (2018).

---

> > > ### Author Response · Authors · 2021-08-30
> > > **Reply to Reviewer RJD5**
> > >
> > > We are glad to hear the overall positive response from the reviewer and thank the reviewer's constructive comments. In the following, we address each issue raised by the reviewer.
> > >
> > >
> > >
> > > >Comment:
> > > >Thank you for your reply, and the edits you have made in response to my feedback.
> > > >
> > > >Except for some additional comments below, I am happy that you have adequately addressed the concerns raised in my review. If the concerns below are also adequately addressed, I will increase my score.
> > > >
> > > >Section 3 Residual2vec graph embedding, Line 91-92
> > > >
> > > >I appreciate that unweighted graphs can be represented as weighted graphs with all weights set to 1. However, I think the text needs to compare the assumed graph type used to calculate the baseline probabilities, and the graph types used in the experiments. Specifically, whether they permit self-loops and integer-weighted/multi- edges. You ought to state that you assume that the baseline probability calculated for graphs with self-loops and integer-weighted/multi- edges is a good approximation for the graph types used in the experiments.
> > > >
> > >
> > > We have added text to clarify the graph types as follows:
> > >
> > > - (line 116):
> > >
> > >    *We assume that the given graph is undirected and weighted, although our results can be generalized to directed graphs. We allow multi-edges (i.e., multiple edges between the same node pair) and self-loops, and consider unweighted graphs as weighted graphs with all edge weight set to one.*
> > >
> > > - (line 174):
> > >
> > >    *The "soft configuration model" allows self-loops and multi-edges---which are not present in the graphs used in the benchmarks---and thus is not perfectly compatible with the benchmark graphs. Nevertheless, because the multi-edges and self-loops are rare in the case of sparse graphs, the soft configuration model has been widely used for sparse graphs without multi-edges and self-loops [a1-a3].*
> > >
> > >
> > > [a1] Newman, M. E. J. 2006. “Finding Community Structure in Networks Using the Eigenvectors of Matrices.” Physical Review E 74 (3): 036104.
> > >
> > > [a2] Newman, M. E. J. 2014. “Networks: An Introduction.” Oxford University, 163–86.
> > >
> > > [a3] Fosdick, B., D. Larremore, J. Nishimura, and J. Ugander. 2018. “Configuring Random Graph Models with Fixed Degree Sequences.” SIAM Review 60 (2): 315–55.
> > >
> > > >Section 3.1 Random graph models and Table 1
> > > >
> > > >I am glad that the authors have clarified that they are preserving expected quantities in their graph models rather than exact quantities.
> > > >
> > > >One point I would like to add is that the term "canonical random graph model" does not seem to be standard in the literature. Instead, the term "canonical ensemble" is usually used, and understood in the context of a constraint (on an average quantity) in the graphs. If you prefer not to introduce the concept of canonical network ensembles, I think it is sufficient to explain that it is the expected quantities that are preserved. I would also suggest clarifications as to which version of the configuration and Erdős-Rényi models are assumed with regards to self-loops and multi-edges.
> > > >
> > > >In the case of preserved expected nodes degrees, I suggest referring to the model as the soft configuration model, with self-loops and multi-edges permitted.
> > > >
> > > >In the case of preserved expected number of edges, I suggest referring to the model as a multigraph version of the Erdős-Rényi random graph model as is done in [b1] (whilst still explaining that the expected number of edges is preserved).
> > > >
> > >
> > > Thank you for pointing this out. We agree that "canonical ensembles" are commonly used in the literature and thus would be a more appropriate term. We also agree that we should clarify the type of random graph models assumed in our paper.
> > > To incorporate these points, we modified the text as follows:
> > >
> > > - (line 112):
> > >
> > >   *The dcSBM can be mapped to many canonical ensembles that preserve the expectation of structural properties. In fact, when $B=1$, the dcSBM is reduced to the soft configuration model that preserves the degree of each node on average, with self-loops and multi-edges allowed. Furthermore, by setting $B = 1$ and $d_i=\text{constant}$, the dcSBM is reduced to the Erdős-Rényi model for multigraphs that preserves the number of edges on average, with self-loops and multi-edges allowed.*
> > >
> > >
> > >
> > > >For the example of three nodes a, b and c:
> > > >
> > > >There are many variants of the configuration model [b2]. If you allow self-loops, then b and c can be connected. Provided the column heading is changed to "soft configuration model", and the text makes it clear that self-loops and multi-edges are assumed in the calculation, I am happy with the baseline probability listed in Table 1.
> > > >
> > > >[b1] Karrer, B. & Newman, M. E. J. Stochastic blockmodels and community structure in networks. Physical Review E 83, 016107. issn: 1539-3755. http://link.aps.org/doi/10.1103/PhysRevE.83.016107 (2011)
> > > >
> > > >[b2] Fosdick, B, Larremore, D, Nishimura, J & Ugander, J. Configuring Random Graph Models with Fixed Degree Sequences. SIAM Review 60, 315–355 (2018).
> > >
> > > We agree. We amended the text as follows:
> > >
> > > - (Table 1):
> > >
> > >   We have replaced "The configuration model" with "The soft configuration model", and "Erdős-Rényi random graph" with "Erdős-Rényi model for multigraphs".
> > >
> > > - (line 124):
> > >
> > >   *This $p_0(j)$ is equivalent to the baseline for the soft configuration model [20]*
> > >
> > > - (line 128):
> > >
> > >   *DeepWalk is equivalent to residual2vec with the Erdős-Rényi model for multigraphs as the null model.*
> > >
> > > - (line 173):
> > >
> > >   *We use the soft configuration model [20] as the null graph for residual2vec, denoted by r2v-config, which yields a degree-debiased embedding.*
> > >
> > >
> > > In summary, we added and modified the text to clarify the type of graphs that we assume, especially whether or not multi-edges and self-loops are allowed.
> > > We deeply appreciate the reviewer for a number of constructive suggestions, which greatly improve our manuscript!

---

> > > > ### Comment · Reviewer_RJD5 · 2021-08-30
> > > > **Reply to Authors**
> > > >
> > > > Thank you for your response. I am happy with the edits that you have made, and satisfied that my comments have been adequately addressed.
> > > >
> > > > Based on your edits, I have edited my rating, increasing it from 4 to 6. I would like to thank the authors for the time they have spent considering my comments and editing their paper.

---

### Official Review · Reviewer_yRQF · 2021-07-17

**Rating:** 8
**Confidence:** 3

**Summary:**

There are two major contribution from this paper: 1) the authors provide the insight that the word2vec trained with SGNS is implicitly canceling out the bias due to the degree of the node; 2) the authors use this insight to introduce a framework that allows removing any structural bias in the graph, so that the true affinity between nodes can be revealed. Under this framework, the authors proposed to use the degree-corrected stochastic block model as the baseline probabilities, and learn the residual information on top of the baseline model. The authors also show that residual2vec can be understood as matrix factorization.

The authors use residual2vec empirically for link prediction and community detection, and find it works empirically the best against competing baselines. Moreover, in the case study, the authors show how residual2vec can be used to remove the bias introduced by publication years, and suggested how the model can be used to remove social bias in many real-world applications.

**Limitations And Societal Impact:**

I find great positive societal impact of this paper. The method proposed in this paper can potentially be used to implicitly remove gender bias, racial bias, and age bias in social graphs.

**Main Review:**

This paper brings a lot of insights to the existing methods such as node2vec and deepwalk, and explains why certain choice of p0 works well in different scenarios. The proposed residual2vec is efficient, gives good empirical performance, and is widely applicable to different scenarios. The writing is super clear and the experiments are very insightful. I haven't follow the node embedding literatures closely, so I will rely on other reviewers to judge the originality.


**Time Spent Reviewing:**

5

---

> ### Author Response · Authors · 2021-08-11
> **Reply to Reviewer yRQF**
>
> We thank the reviewer's careful reading of our paper and are glad to hear the overall positive response.

---

### Official Review · Reviewer_yRN6 · 2021-07-17

**Rating:** 7
**Confidence:** 4

**Summary:**

The paper proposes an unsupervised graph representation learning method, named residual2vec. The random walk-based approaches might be biased in sampling the center-context node pairs due to the network structure and many methods employ a noise distribution to sample the negative instances. The paper investigates the assumptions taken into the consideration during the sampling procedure. The authors formulate the sampling procedure under a general framework, and they show that the existing methodologies can be expressed as special cases of the proposed approach. The performance of the algorithm is evaluated on the link prediction and community detection tasks.

**Ethical Concerns:**

No ethical concerns.

**Limitations And Societal Impact:**

The authors discuss the limitations and possible social impact of the proposed methodology.

**Main Review:**

The organization of the paper is quite good, and it is well-written but some points could be enhanced. For instance, the definitions of the probability distributions can be more explicitly given in order to reduce the ambiguity. The difference between the notations $p_0()$ and $P_0()$ is not clear.

The authors clearly explain the proposed methodology with an illustrative figure. They consider a problem that has not been well studied before. The paper generalizes existing random walk-based approaches by reformulating the sampling procedures and proposes a novel framework. The paper could attract the interest of the community working in the graph representation learning field.

Here, I would like to add some notes and ask questions to clarify some points about the paper.

-  Could the authors elaborate more on what the offset values are, which are used in the link prediction task? And why the offset values for Node2Vec and DeepWalk are not set?
- Why is the parameter B to 1,000? Does the parameter depend on the network size or structure?
-  Why don’t the authors use the NetMF approach in the experimental evaluation? It is also cited in Table 1, and it is very relevant to the proposed approach. It learns the embeddings by factorizing the matrix based on the random walks.
- In the experimental evaluation,  the authors might use more recent approaches in order to evaluate the performance of the proposed methodology.
- The paper proposes a general framework, but it only concentrates on a special case of degree-corrected stochastic block model (dcSBM) with uniform random walk strategy. It might be because of the computational problems but it might be good to examine the influence and the relationship between the chosen random walk distribution and noise distribution. For instance, the authors of the SkipGram approach empirically observe that the model works well for \gamma value equal to 0.75, and similarly, random walk-based approaches apply the same value for the parameter. Why do these models for this common choice show better performance in the downstream tasks?

Typos:

On line 232, Furthremore -> Furthermore
On line 163, computaionally -> computationally


**Time Spent Reviewing:**

5

---

> ### Author Response · Authors · 2021-08-11
> **Reply to Reviewer yRN6**
>
> We would like to thank the reviewer's helpful comments. In the following, we provide detailed responses to all issues raised by the reviewer.
>
> # Clarification of the offset values in link prediction
>
> First, let us highlight that residual2vec decomposes node similarities into two components, i.e., embedding $u_i$ and baseline probability $P_0(j \vert i)$ in Eq. (10). Baseline probability $P_0(j \vert i)$ accounts for the similarities attributed to a null model, and embedding similarity $u_i ^ \top u_j$ represents the "residual" from baseline probability $P_0(j\vert i)$. In the link prediction task, we aimed to leverage both baseline and residual similarities for prediction, by adding offset $z_j = \ln P_0(j \vert i)$ to the embedding similarity $u_i ^\top u_j$. We added $\ln P_0$ instead of $P_0$ by noting that Eq. (10) can be rewritten as
>
> $$
> P_{\text{r2v}}(j \vert i) = \frac{P_0( j \vert i)\exp( u_i ^\top u_j)}{Z_i '} = \frac{\exp( u_i ^\top u_j + \ln P_0( j \vert i))}{Z_i '}.
> $$
>
> In other words, $\ln P_0 (j \vert i)$ has the same unit as the embedding similarity $u_i ^ \top u_j$ in the model. Therefore, we adopted $\ln P_0 (j \vert i)$ as the offset $z_i$.
>
> Glove has a parameter (i.e., bias term) equivalent to $z_i$, which is used for prediction in our benchmark. DeepWalk and Node2Vec do not have the baseline parameters, and thus we set $z_i = 0$.
>
> To clarify this point, we have added text to clarify what the offset values represent in the main text and the Supplementary Information as follows:
>
> * (line 196)
>
>   *We leverage both embedding $u_i$ and baseline probability $P_0(j \vert i)$ to predict missing edges. Specifically, we calculate the prediction score by $u_i ^\top u_j + z_i + z_j$, where we set $z_j=\ln P_0(j \vert i)$ for residual2vec because $\ln P_0(j \vert i)$ has the same unit as $u_i ^\top u_j$ (Supplementary Information). Glove has a bias term that is equivalent to $z_i$. Therefore, we set $z_i$ to the bias term for Glove. Other methods do not have the parameter that corresponds to $z_i$ and thus we set $z_i = 0$.*
>
>
> # Choice of $\hat B$
>
> We appreciate your question, which gives us the opportunity to clarify our parameter choice.
> Parameter $\hat B$ determines the trade-off between the speed and the accuracy of the approximation by the dcSBM. We acknowledge that the appropriate choice of $\hat B$ might depend on the size and structure of graphs.
> Thus, it is a hyperparameter that can be tuned for given resource constraints.
> In our case, we wanted to make the method as efficient as possible while maintaining a decent approximation accuracy.
> Therefore, we tested our approximation using the six empirical graphs from different domains and confirmed that setting $\hat B = 1000$ yielded a high approximation accuracy for all the graphs in terms of the Pearson correlation coefficient (i.e, 0.85 on average; Figure 1 in the Supplementary Information) while achieving good scalability.
>
> # Adding NetMF as a baseline
>
> This is a good point. We have added NetMF as a baseline in the benchmarks.
> The AUC-ROC values for NetMF are, on average, 54% and 98% of those for residual2vec in the link prediction and community detection benchmarks, respectively.
> NetMF performed especially well in the community detection, i.e, it achieved the fifth highest performance followed by the three node2vecs with different parameter configurations ($q=1, 0.5, 2$).
> Nevertheless, NetMF did not outperform residual2vec in both benchmarks. For the WoS citation graph, we could not run NetMF because NetMF exceeded our memory capacity (1Tb RAM) when it created the DeepWalk matrix, which is a dense $N\times N$ matrix, where $N$ is the number of nodes (see the NetMF paper [a1]).
>
> # Comparison with recent approaches
>
> While we acknowledge that comparing our method with more recent approaches is useful, we believe that it will also make our point more obscure. Because our method and contribution heavily focus on the statistical biases in the random walks and their manifestation in graph embedding results, we believe that the most appropriate informative context is the class of methods based on the random walks.
> Therefore, we would like to leave the in-depth comparison with other recent approaches for future work, although we have added several methods (Fairwalk [a2] and GAT [a3]) to the benchmarks in this revision and confirmed that our message has not changed.
>
> # Noise distribution and random walks
>
> Thank you for the interesting question. The role of $\gamma$ on the downstream performance is indeed a fascinating question---which we do not know how to answer.
> $\gamma\neq 1$ makes it much more difficult to analytically approach the problem, and thus it is still an open question for us. We hope future research can shed more light on this.
>
> # Typos
>
> Thank you for spotting them. We have fixed the typos.
>
>
> In summary, we have updated our manuscript by clarifying the offset values in the link prediction and our parameter choice for $\hat B$, and by adding NetMF as a baseline.
> We thank once again the reviewer for constructive comments and thoughts!
>
>
> [a1] Qiu, Jiezhong, Yuxiao Dong, Hao Ma, Jian Li, Kuansan Wang, and Jie Tang. 2018. “Network Embedding as Matrix Factorization: Unifying DeepWalk, LINE, PTE, and Node2vec.” Proceedings of the Eleventh ACM International Conference on Web Search and Data Mining, WSDM ’18, 2018-Febua: 459–67.
>
> [a2] Rahman, Tahleen, Bartlomiej Surma, Michael Backes, and Yang Zhang. 2019. “Fairwalk: Towards Fair Graph Embedding.” In Proceedings of the Twenty-Eighth International Joint Conference on Artificial Intelligence, 3289–95. California: International Joint Conferences on Artificial Intelligence Organization.
>
> [a3] Veličković, Petar, Guillem Cucurull, Arantxa Casanova, Adriana Romero, Pietro Liò, and Yoshua Bengio. 2018. “Graph Attention Networks.” In International Conference on Learning Representations.

---

> > ### Comment · Reviewer_yRN6 · 2021-08-31
> > **Comment for the response**
> >
> > I would like to thank the authors for their response. I have increased my score (7: Good paper, accept). Overall, the paper makes a good contribution. Still, the paper could be enhanced with the addition of a more diverse set of baseline models.

---

### Decision · Program_Chairs · 2021-09-28

**Decision:**

Accept (Poster)

**Comment:**

The paper presents Residual2Vec, a technique for reducing structural biases for graph embedding. Existing methods use random walk-based sampling, which has a strong preference towards "hub" nodes. The paper draws an insightful analogy of how debasing effectively happens in negative sampling in word2vec, which inspired the authors to develop Residual2Vec to compensate structural biases in random walks. The reviewers are overall happy with the paper, with questions and concerns mostly regarding the presentation of the work. These questions are mostly clarified during the rebuttal, which the authors should include in the final version of the paper. The work makes a useful contribution for learning effective graph node embeddings with less dependency on node degrees.

**Consistency Experiment:**

NeurIPS has a long history of experimentation. In 2014, NeurIPS ran an experiment in which 10% of submissions were reviewed by two independent committees to quantify the randomness in the review process. This year, we repeated a variant of this experiment to see how the quality of the review process has changed over time.  This paper was part of the experiment and was therefore assigned to two committees (consisting of reviewers, an Area Chair, and a Senior Area Chair) that reached independent decisions.  If both committees made the same recommendation, this recommendation was followed. If a single committee recommended acceptance, the paper was accepted (with the exception of a few cases in which the other committee identified what we considered a fatal flaw, e.g., an error in a key result).

This copy’s committee reached the following decision: **Accept (Spotlight)**

The other committee assigned to the paper recommended **Reject**.  You can find the other set of reviews, along with any follow up discussion with the authors here:
https://openreview.net/forum?id=Z9K7sds_-jC